# Prior Forgetting and In-Context Overfitting

**Sungyoon Lee**
Department of Computer Science
Hanyang University
sungyoonlee@hanyang.ac.kr

## Abstract

In-context learning (ICL) is one of the key capabilities contributing to the great success of LLMs. At test time, ICL is known to operate in the two modes: task recognition and task learning. In this paper, we investigate the emergence and dynamics of the two modes of ICL during pretraining. To provide an analytical understanding of the learning dynamics of the ICL abilities, we investigate the in-context random linear regression problem with a simple linear-attention-based transformer, and define and disentangle the strengths of the task recognition and task learning abilities stored in the transformer model's parameters. We show that, during the pretraining phase, the model first learns the task learning and the task recognition abilities together in the beginning, but it (a) gradually forgets the task recognition ability to recall the priorly learned tasks and (b) relies more on the given context in the later phase, which we call (a) *prior forgetting* and (b) *in-context overfitting*, respectively.

## 1 Introduction

Large language models (LLMs) show an emergent behavior [Wei et al., 2022], known as in-context learning (ICL) [Brown et al., 2020, Kaplan et al., 2020], that they can learn (without updating any model parameters) a new unseen task from a few demonstrations of input-output pairs given at test time. While models trained with traditional supervised learning for a given task/mapping learn task-specific features that might be spurious or irrelevant for other tasks, the ICL ability enables the models to learn stronger representations with task-agnostic architecture and task-agnostic data and to efficiently learn diverse tasks via text interaction in a flexible way. Thus, it is often considered as one of the key capabilities contributing to the great success of LLMs.

At test time, in-context learning is known to operate in the following two modes: given an in-context task, the model (i) recalls similar functions and concepts learned (priorly) in the pretraining phase and (ii) smoothly adapts to and implicitly learns the (observed) in-context task [Xie et al., 2022, Raventós et al., 2024, Pan et al., 2023, Lin and Lee, 2024]. These dual Bayesian modes of ICL are often called (i) *task recognition* and (ii) *task learning* [Pan et al., 2023]. In other words, the pretrained model (i) knows the prior task distribution and (ii) has the ability to select and perform a proper task corresponding to the given demonstrations.

The power of each mode of ICL is not the same and one often dominates the other depending on many factors. To separately investigate the effects of the task recognition and task learning abilities, it has been proposed to use the experimental setups with noisy output labels (random or semantically irrelevant labels). Min et al. [2022], Lyu et al. [2023] show that randomly replacing output labels in the demonstrations does not significantly affect ICL performance which indicates that the task recognition dominates the task learning. On the other hands, Yoo et al. [2022], Wei et al. [2023], Shi et al. [2023] show that larger models are easily distracted by wrong or irrelevant demonstrations and Pan et al. [2023] show that larger models exhibits a better task learning ability than smaller ones and

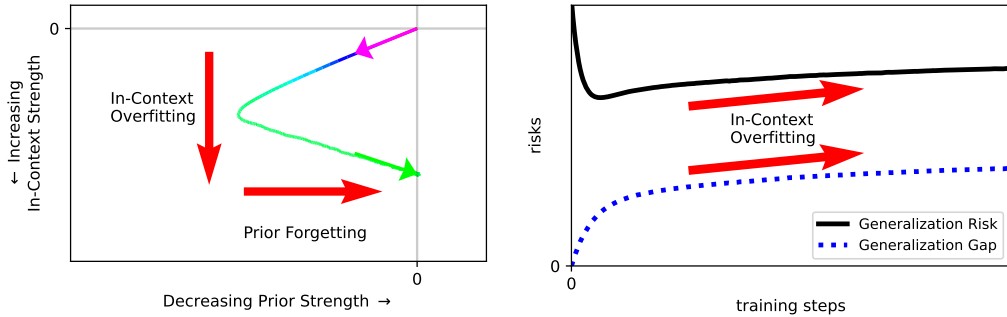

Figure 1: **Conceptual sketch of in-context overfitting and prior forgetting.** (Left) Following the gradient flow (the c-shaped trajectory ($\varsigma$) from magenta to lime) in the parameter space, we observe a monotonic increase in *in-context strength* ($\downarrow$) during training, which we call **in-context overfitting**, and a decrease in *prior strength* ($\rightarrow$) in the later phase of the training, which we call **prior forgetting**. The exact meaning of the axes will become clear in Figure 2 (Bottom). (Right) We observe a monotonic increase in the *generalization gap* (blue dotted line) and a u-shaped curve of the *generalization risk* (black solid line). We also call these two phenomena **in-context overfitting** ($\nearrow$) since the generalization gap is directly linked with the in-context strength.

it improves with more demonstrations. In other words, the task learning ability emerges as we scale up the model and it plays more important role than the task recognition for larger models.

As shown above, most studies attempt to analyze the behaviors of *the pretrained transformers at test time*, but to deeper understand the two modes of ICL, we need to investigate their emergence and dynamics *in the pretraining phase*. Thus, we raise the following questions:

> *How do in-context learning abilities emerge (and disappear) during pretraining?*

To answer this question with an analytical understanding of the learning dynamics of the ICL abilities, we investigate the in-context random linear regression problem [Garg et al., 2022, Akyürek et al., 2023, Von Oswald et al., 2023a, Li et al., 2023b] with a simple linear-attention-based transformer [Schlag et al., 2021, Von Oswald et al., 2023a]. Moreover, to mathematically model a setup similar to the experimental settings considered in Min et al. [2022], Lyu et al. [2023], Yoo et al. [2022], Wei et al. [2023], Shi et al. [2023], Pan et al. [2023] and to disentangle the strengths of the task recognition and task learning abilities, we introduce new settings we call *demonstration-query task irrelevance and noncentral task model*. With these small modifications from previous works, we can derive a simple yet interesting dynamics regarding the ICL abilities.

To be specific, we show that, during the pretraining phase, the model first learns the task learning and task recognition abilities together in the beginning, but it (a) gradually forgets the task recognition ability to recall the priorly learned tasks and (b) relies more on the given context in the later phase, which we call (a) *prior forgetting* and (b) *in-context overfitting*, respectively. Figure 1 illustrates prior forgetting and in-context overfitting from the perspectives of model parameters (Left) and generalization loss (Right).

## 2 Related Work

After it was demonstrated that large transformers [Vaswani et al., 2017] such as GPT-3 can perform in-context learning [Brown et al., 2020, Kaplan et al., 2020], there has been a growing interest in understanding the underlying mechanisms of the ICL ability.

Garg et al. [2022] empirically show that, for in-context linear regression, the trained transformers match the performance of the optimal least squares estimator, i.e., the solution obtained by gradient flow on the in-context examples. This work is followed by Von Oswald et al. [2023a], Akyürek et al. [2023] in which the authors provide a construction of transformer that can implement gradient descent for in-context linear regression. Especially, the construction in Von Oswald et al. [2023a] only requires a single linear self-attention layer [Schlag et al., 2021] to implement a single iteration

of gradient descent. After that, many works consider linear transformer as a simple proxy for the softmax-based transformers to theoretically investigate their complex behaviors [Mahankali et al., 2024, Ahn et al., 2024a,b, Zhang et al., 2024].

Ahn et al. [2024a] show that, for a certain training objective over random instances of linear regression, the global minimizers can implement preconditioned gradient descent. Here, the parameter configuration of the minimizer is the same (up to a constant factor) as the one constructed in Von Oswald et al. [2023a]. Moreover, Zhang et al. [2024] show that such global minimizers can be achieved by running gradient flow under a certain initialization inducing the balancedness condition [Arora et al., 2019a, 2018, 2019b, Du et al., 2018] throughout the training.

Beyond the linear regression, Garg et al. [2022] also explore more complex function classes such as two-layer neural networks and decision trees, and empirically show that the trained transformer can in-context learn these function classes. Some theoretical work extend the in-context linear regression problem to exponential regression [Gao et al., 2023], softmax regression [Li et al., 2023a] or autoregressive learning [Von Oswald et al., 2023b, Sander et al., 2024, Zheng et al., 2024]. Moreover, Dai et al. [2023] empirically explore language models on real NLP tasks such as sentiment classification, topic classification, and natural language inference.

Likewise, many studies attempt to analyze the behaviors of the *pretrained* transformers at test time, e.g., the two Bayesian modes of ICL [Xie et al., 2022, Wies et al., 2023, Jiang, 2023, Wang et al., 2024b, Raventós et al., 2024, Pan et al., 2023, Lin and Lee, 2024] and their relative performance (especially when we scale the model) [Min et al., 2022, Lyu et al., 2023, Yoo et al., 2022, Wei et al., 2023, Shi et al., 2023, Pan et al., 2023, Shi et al., 2024]. On the other hand, we focus more on the emergence and rise-and-decline dynamics of the test-time performance of the two modes *during pretraining*.

To model the task distribution, Raventós et al. [2024], Lin and Lee [2024] introduce a probabilistic mixture model of multiple task groups with task-dependent input distributions, while we analyze a simple unimodal task distribution.

There are some more interesting work on the power balance between the two Bayesian modes of ICL. Wang et al. [2023] show a similar results with Min et al. [2022] for chain-of-thought prompting that invalid reasoning steps do not hurt performance on multi-step reasoning tasks. Reynolds and McDonell [2021] show that zero-shot prompts can match and even outperform few-shot prompts, which implies that the task recognition ability plays a more important role than from the task learning ability for some tasks.

Wang et al. [2024a] design a metric called competition intensity to explore the emergence of ICL and empirically show that the two modes of ICL are competitive during pretraining, while we measure the strengths of the two modes from the transformer's parameters and theoretically investigate how they emerge and disappear.

## 3 Settings

In this section, we first investigate the in-context random linear regression problem with a simple linear-attention-based transformer, following [Garg et al., 2022, Akyürek et al., 2023, Von Oswald et al., 2023a, Li et al., 2023b, Schlag et al., 2021], in Section 3.1. Then, we introduce some new settings and definitions to explore robustness to shift that we are given demonstrations irrelevant to the query task as similar to the empirical settings considered in Min et al. [2022], Lyu et al. [2023], Yoo et al. [2022], Wei et al. [2023], Shi et al. [2023], Pan et al. [2023] in Sections 3.2–3.3. See Appendix A for a quick reference for the notations.

### 3.1 In-Context Linear Regression with Linear Transformer

We train a transformer with the training set, which consists of the input context matrices and the corresponding target responses. The input context matrix

$$Z = \begin{bmatrix} X \\ Y \end{bmatrix} = \begin{bmatrix} \bar{X} & \boldsymbol{x}^{(n+1)} \\ \bar{Y} & 0 \end{bmatrix} = \begin{bmatrix} \boldsymbol{x}^{(1)} & \boldsymbol{x}^{(2)} & \cdots & \boldsymbol{x}^{(n)} & \boldsymbol{x}^{(n+1)} \\ y^{(1)} & y^{(2)} & \cdots & y^{(n)} & 0 \end{bmatrix} \in \mathbb{R}^{(d+1)\times(n+1)}$$

is generated by drawing $n + 1$ $d$-dimensional covariates $\boldsymbol{x}^{(i)}$ and an in-context task vector $\boldsymbol{w}$ representing a linear function $f_{\boldsymbol{w}} : \boldsymbol{x} \mapsto \boldsymbol{w}^\top \boldsymbol{x}$ and computing the target responses $y^{(i)}$ as follows:

$$\boldsymbol{x}^{(i)} \stackrel{\text{i.i.d.}}{\sim} \mathcal{D}_{\mathcal{X}}, \boldsymbol{w} \sim \mathcal{D}_{\mathcal{W}}, y^{(i)} = \boldsymbol{w}^\top \boldsymbol{x}^{(i)} \ (i = 1, \cdots, n + 1),$$

where $\boldsymbol{x}^{(i)}, \boldsymbol{w} \in \mathbb{R}^d$, $y^{(i)} \in \mathbb{R}$, $X = [\bar{X} \quad \boldsymbol{x}^{(n+1)}] \in \mathbb{R}^{d \times (n+1)}, \bar{X} = [\boldsymbol{x}^{(1)} \cdots \boldsymbol{x}^{(n)}] \in \mathbb{R}^{d \times n}$, $Y = [\bar{Y} \quad 0] \in \mathbb{R}^{1 \times (n+1)}, \bar{Y} = [y^{(1)} \cdots y^{(n)}] \in \mathbb{R}^{1 \times n}$. Here, the $\boldsymbol{x}^{(i)}$'s for $i \leq n$ and $\boldsymbol{x}^{(n+1)}$ are called the in-context covariates and the query input, respectively.

We consider a single-layer linear transformer $T_{P,Q}$ with linear self-attention (LSA) [Schlag et al., 2021, Von Oswald et al., 2023a] parametrized by two matrices $P$ and $Q$ and residual connection [He et al., 2016]:

$$T_{P,Q}(Z) = - \left[ Z + \frac{1}{n} \text{LSA}_{P,Q}(Z) \right]_{d+1, n+1}, \tag{1}$$

$$\text{LSA}_{P,Q}(Z) = PZMZ^\top QZ,$$

where

$$P = \begin{bmatrix} \mathbf{0}_{d \times d} & \mathbf{0}_d \\ \boldsymbol{p}^\top & \kappa \end{bmatrix}, \ Q = \begin{bmatrix} \bar{Q} & \mathbf{0}_d \\ \boldsymbol{q}^\top & 0 \end{bmatrix} \in \mathbb{R}^{(d+1) \times (d+1)}, \ M = \begin{bmatrix} I_n & \mathbf{0}_n \\ \mathbf{0}_n^\top & 0 \end{bmatrix} \in \mathbb{R}^{(n+1) \times (n+1)}.$$

Here, the first $d$ rows $P_{1:d,:}$ of $P$ and the last column $Q_{:,d+1}$ of $Q$ do not affect the output $T_{P,Q}(Z)$ of the transformer (see Appendix B), i.e., during the gradient-based training they remain the same as the initial values which are usually very small, and thus we simply put $P_{1:d,:} = \mathbf{0}_{d \times (d+1)}$ and $Q_{:,d+1} = \mathbf{0}_{d+1}$.

Note that the usual self-attention (SA) can be expressed as

$$\text{SA}_{W_Q, W_K, W_V; \sigma}(Z) = W_V ZM\sigma(Z^\top W_K^\top W_Q Z),$$

where $\sigma$ is usually the column-wise softmax [Bahdanau et al., 2015, Vaswani et al., 2017]. Here, the linear self-attention is a special case of the self-attention when $P = W_V, Q = W_K^\top W_Q, \sigma(Z) = Z$.

We say the transformer *performs a task* $\hat{\boldsymbol{w}}$ if $T_{P,Q}(Z) = f_{\hat{\boldsymbol{w}}}(\boldsymbol{x}^{(n+1)}) = \hat{\boldsymbol{w}}^\top \boldsymbol{x}^{(n+1)}$. For the parametrization in (1), we can express the task

$$\hat{\boldsymbol{w}} = -(\bar{Q}^\top + \boldsymbol{q}\boldsymbol{w}^\top) G_x (\boldsymbol{p} + \kappa \boldsymbol{w}), \tag{2}$$

where $G_x = \frac{1}{n} \bar{X} \bar{X}^\top = \frac{1}{n} \sum_{i=1}^n \boldsymbol{x}^{(i)} \boldsymbol{x}^{(i)\top}$. See Appendix B for details.

To learn to predict the target response $y^{(n+1)} = \boldsymbol{w}^\top \boldsymbol{x}^{(n+1)}$ for the query input, the training loss (also called the training risk) is given as

$$L_{\text{train}}(P, Q) \equiv \mathbb{E}_{\boldsymbol{w}, X} \left[ \left( \boldsymbol{w}^\top \boldsymbol{x}^{(n+1)} - T_{P,Q}(Z) \right)^2 \right] = \mathbb{E}_{\boldsymbol{w}, X} \left[ \left( (\boldsymbol{w} - \hat{\boldsymbol{w}})^\top \boldsymbol{x}^{(n+1)} \right)^2 \right].$$

## 3.2 Demonstration-Query Task Irrelevance and Noncentral Task Model

Zhang et al. [2024] consider the following three distribution shifts:

- **task shifts**: the tasks provided in the pretraining phase and the tasks at test time follow different distributions.
- **query shifts**: the in-context covariates $\boldsymbol{x}_{\text{test}}^{(i)}$ and the query input $\boldsymbol{x}_{\text{test}}^{(n+1)}$ follow different distributions.
- **covariate shifts**: the in-context inputs $X$ in the pretraining phase and the test phase follow different distributions.

We depart from these distribution shifts and the vanilla setup (without any distribution shifts) considered in previous work, and introduce another class of scenario, which we call *demonstration-query task irrelevance*, where the demonstration task $\boldsymbol{w}$ and the query task $\boldsymbol{w_q}$ (both at test time) are different (independent) but sampled from the same distribution:

**Assumption 3.1** (Demonstration-Query Task Irrelevance).

$$\boldsymbol{w}, \boldsymbol{w_q} \overset{\text{i.i.d.}}{\sim} \mathcal{D}_{\mathcal{W}}.$$

In other words, we are given an input context matrix

$$Z_{\text{test}} = \begin{bmatrix} X_{\text{test}} \\ Y_{\text{test}} \end{bmatrix} = \begin{bmatrix} \boldsymbol{x}_{\text{test}}^{(1)} & \cdots & \boldsymbol{x}_{\text{test}}^{(n)} & \boldsymbol{x}_{\text{test}}^{(n+1)} \\ \boldsymbol{w}^{\top}\boldsymbol{x}_{\text{test}}^{(1)} & \cdots & \boldsymbol{w}^{\top}\boldsymbol{x}_{\text{test}}^{(n)} & 0 \end{bmatrix}$$

with the demonstration task $\boldsymbol{w}$, but the target response is determined by another independent query task $\boldsymbol{w_q}$, not by the demonstration task $\boldsymbol{w}$:

$$y_{\text{test}}^{(n+1)} = \boldsymbol{w_q}^{\top}\boldsymbol{x}_{\text{test}}^{(n+1)}.$$

Note that this assumption is designed not to model a real-world scenario but to disentangle the power of the two ICL abilities because it is similar to the empirical settings considered in Min et al. [2022], Lyu et al. [2023], Yoo et al. [2022], Wei et al. [2023], Shi et al. [2023], Pan et al. [2023] that we are given demonstrations irrelevant to the query task.

To investigate the robustness and generalization to the demonstration-query irrelevance, we consider the following generalization risk (also called test risk):

$$L_{\text{test}}(P, Q) \equiv \mathbb{E}_{\boldsymbol{w_q}, \boldsymbol{w}, X_{\text{test}}} \left[ \left( \boldsymbol{w_q}^{\top}\boldsymbol{x}_{\text{test}}^{(n+1)} - T_{P,Q}(Z_{\text{test}}) \right)^2 \right]$$

and the generalization gap between the training risk and the generalization risk defined as follows:

$$\Delta L(P, Q) \equiv L_{\text{test}}(P, Q) - L_{\text{train}}(P, Q). \tag{3}$$

If the task learning plays a more important role than the task recognition, then the model relies more on the demonstration task that is irrelevant to the query task and thus it shows a higher generalization risk and gap. Therefore, by measuring the generalization risk or gap, we can separately analyze the task learning ability from the task recognition ability.

Similarly, we also want to separately investigate the power of the task recognition ability with which, given a demonstration task, the model can recall similar concepts from prior knowledge to infer the query task. However, the two tasks are independent. Therefore, we need to consider *shared concept* in the prior knowledge between the two tasks.

To this end, we simply examine a *non-centeral* task distribution $\mathcal{D}_{\mathcal{W}}$ with a nonzero mean $\boldsymbol{\mu} \neq \boldsymbol{0}_d$. Here, the task center $\boldsymbol{\mu}$ explains the prior task distribution in the sense that $\boldsymbol{w} = \boldsymbol{\mu} + \boldsymbol{s}$ and $\boldsymbol{w_q} = \boldsymbol{\mu} + \boldsymbol{s_q}$ share the (non-zero) prior knowledge $\boldsymbol{\mu}$ and they have their own knowledge $\boldsymbol{s}$ and $\boldsymbol{s_q}$. By using this prior knowledge represented by a nonzero vector, we can measure how much the model knows and utilizes this prior, i.e., the power of the task recognition ability (see (7)).

**Assumption 3.2** (Isotropic Covariate and Noncentral Task). We assume that

   (i) the covariate distribution is $\mathcal{D}_{\mathcal{X}} = \mathcal{N}(\boldsymbol{0}_d, I_d)$ and

   (ii) the task distribution $\mathcal{D}_{\mathcal{W}} = \mathcal{N}(\boldsymbol{\mu}, \Sigma_{\mathcal{W}})$ is noncentral with $\|\boldsymbol{\mu}\| = 1$ and isotropic with $\Sigma_{\mathcal{W}} = \sigma^2 I_d$.

Here, we name $b \equiv \text{tr}(\Sigma_{\mathcal{W}}) = \sigma^2 d$ *the task dispersion*.

Under the above isotropic covariate assumption (i), we have

$$L_{\text{train}}(P, Q) = \mathbb{E}_{\boldsymbol{w}, X} \left[ \left( (\boldsymbol{w} - \hat{\boldsymbol{w}})^{\top}\boldsymbol{x}^{(n+1)} \right)^2 \right] = \mathbb{E}_{\boldsymbol{w}, \bar{X}}[\|\boldsymbol{w} - \hat{\boldsymbol{w}}\|^2] \tag{4}$$

$$L_{\text{test}}(P, Q) = \mathbb{E}_{\boldsymbol{w_q}, \boldsymbol{w}, X} \left[ \left( (\boldsymbol{w_q} - \hat{\boldsymbol{w}})^{\top}\boldsymbol{x}^{(n+1)} \right)^2 \right] = \mathbb{E}_{\boldsymbol{w_q}, \boldsymbol{w}, \bar{X}}[\|\boldsymbol{w_q} - \hat{\boldsymbol{w}}\|^2]. \tag{5}$$

Note that we assume the isotropic covariate assumption (i) for simplicity which can be easily relaxed to obtain similar equations with (4) and (5) up to a constant multiple, since $\mathbb{E}\left[ \left( \boldsymbol{a}^{\top}\boldsymbol{x}^{(n+1)} \right)^2 \right] = \text{tr}(\Sigma_{\mathcal{X}})\mathbb{E}\left[ \|\boldsymbol{a}\|^2 \right]$ if the covariate has zero-mean and covariance of $\Sigma_{\mathcal{X}}$ instead of $I_d$.

Moreover, with the noncentral task assumption (ii), we define the following generalization risks:

**Definition 3.3** (Generalization Risk at Zero). If the transformer performs the zero-task $\hat{\boldsymbol{w}} = \boldsymbol{0}_d$, i.e., $T_{P,Q}(\boldsymbol{x}^{(n+1)}) = 0$, then the corresponding generalization risk is called *the generalization risk at zero* defined as

$$L_0 \equiv \mathbb{E}_{\boldsymbol{w}_q}\left[\|\boldsymbol{w}_q\|^2\right] = \|\boldsymbol{\mu}\|^2 + \mathrm{tr}(\Sigma_{\mathcal{W}}) = 1 + b.$$

When the parameters are near the small initialization, the generalization risk is about $L_0$.

**Definition 3.4** (Generalization Risk at Random). If the transformer performs a random task $\boldsymbol{w}' \sim \mathcal{D}_{\mathcal{W}}$, i.e., $T_{P,Q}(\boldsymbol{x}^{(n+1)}) = \boldsymbol{w}'^{\top}\boldsymbol{x}^{(n+1)}$, then the expected generalization risk over $\boldsymbol{w}' \sim \mathcal{D}_{\mathcal{W}}$ is called *the generalization risk at random* defined as

$$L_r \equiv \mathbb{E}_{\boldsymbol{w}_q, \boldsymbol{w}'}\left[\|\boldsymbol{w}_q - \boldsymbol{w}'\|^2\right] = 2\,\mathrm{tr}(\Sigma_{\mathcal{W}}) = 2b.$$

**Definition 3.5** (Generalization Risk at Optimum). When the transformer in (1) is optimally trained with the parameter $(P^*, Q^*)$ that minimizes the training loss $L_{\text{train}}$, the train/test risks are called *the train/test risks at optimum* defined as

$$L_{\text{train}}^* \equiv L_{\text{train}}(P^*, Q^*), \quad L_{\text{test}}^* \equiv L_{\text{test}}(P^*, Q^*),$$

and the gap between the two is denoted by $\Delta L^* \equiv L_{\text{test}}^* - L_{\text{train}}^*$. Note that $L_{\text{test}}^*$ is not optimal generalization risk.

### 3.3 Two-Parameter Transformer and Prior/In-Context Strength

Given $\boldsymbol{\mu} \in \mathbb{R}^d$, we consider the following simple transformers parametrization with two scalars $\alpha$ and $\kappa$:

$$P = \begin{bmatrix} \boldsymbol{0}_{d \times d} & \boldsymbol{0}_d \\ \alpha\boldsymbol{\mu}^{\top} & \kappa \end{bmatrix}, Q = \begin{bmatrix} I_d & \boldsymbol{0}_d \\ \boldsymbol{0}_d^{\top} & 0 \end{bmatrix} \in \mathbb{R}^{(d+1) \times (d+1)} \tag{6}$$

which, if we put $\boldsymbol{\mu} = \boldsymbol{0}_d$, reduces to the parameter construction in Von Oswald et al. [2023a] to implement a single step of gradient descent. From now on, we will use the two-parameter notation $T_{\boldsymbol{\theta}}$ with $\boldsymbol{\theta} = [\alpha, \kappa]^{\top}$ instead of $T_{(P,Q)}$.

The two-parameter transformer $T_{\boldsymbol{\theta}}$ *performs* the following task $\hat{\boldsymbol{w}}$, i.e., $T_{\boldsymbol{\theta}}(\boldsymbol{x}^{(n+1)}) = \hat{\boldsymbol{w}}^{\top}\boldsymbol{x}^{(n+1)}$:

$$\hat{\boldsymbol{w}} = -G_x(\alpha\boldsymbol{\mu} + \kappa\boldsymbol{w}), \tag{7}$$

$$-\alpha : \text{prior strength}, \ -\kappa : \text{in-context strength}.$$

Here $\alpha$ and $\kappa$ are the weights of the task center $\boldsymbol{\mu}$ (independent of the demonstrations) and the in-context task $\boldsymbol{w}$, respectively. Thus, we refer to $-\alpha$ and $-\kappa$ (or just $\alpha$ and $\kappa$) as *the (semantic/task) prior strength* and *the in-context strength*, respectively. For example, if $\alpha$ is much larger than $\kappa$ in magnitude, then $\hat{\boldsymbol{w}}$ ignores the demonstrations from $\boldsymbol{w}$, relying heavily on the task prior $\mathcal{D}_{\mathcal{W}}$ (task recognition) represented by $\boldsymbol{\mu}$. On the other hand, if $\kappa$ is much larger than $\alpha$ in magnitude, then $\hat{\boldsymbol{w}}$ only relies on the demonstration task $\boldsymbol{w}$ (task learning) and the model cannot recover irrelevant query task.

This two-parameter model enables us to disentangle the two modes of ICL, the task recognition and task learning abilities, with the two distinct parameters, the prior strength $\alpha$ and in-context strength $\kappa$, respectively. Thus, in the next section, we will explore the evolution of these two parameters.

## 4 Main Results

In this section, first we show that our objective function is quadratic (Theorem 4.1) and then, for this quadratic loss, the gradient flow is a linear ODE which shows a simple dynamics (Theorem 4.2). Using this dynamics, we can analyze how the prior/in-context strengths and generalization gap/risk evolve during pretraining. Moreover, we also discuss how the value of task dispersion $b$ affects the learning dynamics (Theorem 4.3).

### 4.1 Training Dynamics

The training loss function for the two-parameter transformer, from (4) and (7), can be expressed as follows:

$$L_{\text{train}}(\boldsymbol{\theta}) = \mathbb{E}_{\boldsymbol{w}, \bar{X}}[\|\boldsymbol{w} - \hat{\boldsymbol{w}}\|^2] = \mathbb{E}_{\boldsymbol{w}, \bar{X}}[\|\boldsymbol{w} + G_x(\alpha\boldsymbol{\mu} + \kappa\boldsymbol{w})\|^2].$$

From this equation, it can be easily shown that the training loss function is quadratic with respect to the two variables $\alpha$ and $\kappa$. The following theorem states the details of this quadratic loss function:

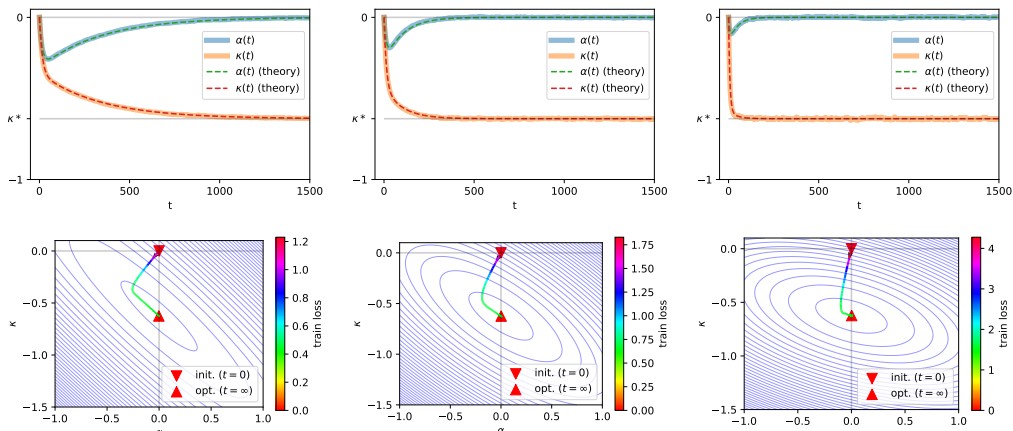

Figure 2: **Evolution of the two parameters, $\alpha$ and $\kappa$, for different $b$'s.** We train the two-parameter transformer using SGD with learning rate of 0.01 and batch size of 4,000. We also use $n = 10, d = 5$, and $\sigma = 0.2, 0.4, 0.8$, i.e., the task dispersion $b = \sigma^2 d = 0.2, 0.8, 3.2$ (from Left to Right). Top: Empirical results with SGD (solid lines) and theoretical results of (11) and (12) with gradient flow (dashed lines). Bottom: Training trajectory (from initialization ▼ to the optimum ▲) drawn with quadratic loss landscape (elliptic level sets) on the $\alpha\kappa$-plane. The color of trajectory indicates the training loss value. See Figure 1 (Left) together.

**Theorem 4.1** (Quadratic Loss). *For the two-parameter transformer, the training loss is quadratic with respect to the parameter $\boldsymbol{\theta} = [\alpha, \kappa]^\top \in \mathbb{R}^2$:*

$$L_{train}(\boldsymbol{\theta}) = \frac{1}{2}\boldsymbol{\theta}^\top C_2 \boldsymbol{\theta} + C_1^\top \boldsymbol{\theta} + C_0, \text{ where} \tag{8}$$

$$C_2 = 2\frac{n+d+1}{n}\begin{bmatrix} 1 & 1 \\ 1 & 1+b \end{bmatrix} \in \mathbb{R}^{2\times 2}, C_1 = 2\begin{bmatrix} 1 \\ 1+b \end{bmatrix} \in \mathbb{R}^2, C_0 = 1 + b \in \mathbb{R},$$

*and $b = \mathrm{tr}(\Sigma_{\mathcal{W}}) = \sigma^2 d$. The training loss $L_{train}(\boldsymbol{\theta})$ is minimized at $\boldsymbol{\theta} = \boldsymbol{\theta}^*$, where*

$$\boldsymbol{\theta}^* = \begin{bmatrix} \alpha^* \\ \kappa^* \end{bmatrix} = \begin{bmatrix} 0 \\ -\frac{n}{n+d+1} \end{bmatrix} \text{ and } L_{train}^* = \frac{d+1}{n+d+1}L_0. \tag{9}$$

Note that, when $\boldsymbol{\mu} = \mathbf{0}_d$, the global minimum $\theta^*$ reduces to the one in Ahn et al. [2024a] as a special case. The proof is deferred to Appendix D.

Figure 2 (Bottom) shows the quadratic loss landscape with elliptic level sets and a unique minimum (since the Hessian matrix $C_2$ is positive-definite) at $\boldsymbol{\theta}^*$ on the $\kappa$-axis (shown with ▲). The geometry of the loss landscape highly affects the learning dynamics as will be detailed in the following theorem and the later sections.

**Theorem 4.2** (Training Dynamics). *Under the same setting of Theorem 4.1, by solving the following linear differential equation (gradient flow) starting from $\boldsymbol{\theta}(0) = 0$:*

$$\dot{\boldsymbol{\theta}} = -\nabla_{\boldsymbol{\theta}} L_{train}(\boldsymbol{\theta}) = -C_2\boldsymbol{\theta} - C_1,$$

*we can obtain the following solution:*

$$\boldsymbol{\theta}(t) = ce^{-\lambda_+ t}\boldsymbol{v}_+ - ce^{-\lambda_- t}\boldsymbol{v}_- + \boldsymbol{\theta}^*, \tag{10}$$

$$c = \frac{n}{(n+d+1)\sqrt{4+b^2}},$$

*where*

$$\lambda_\pm = \frac{2 + b \pm \sqrt{4+b^2}}{2} \text{ and } \boldsymbol{v}_\pm = \left[1, \ \frac{b \pm \sqrt{4+b^2}}{2}\right]^\top$$

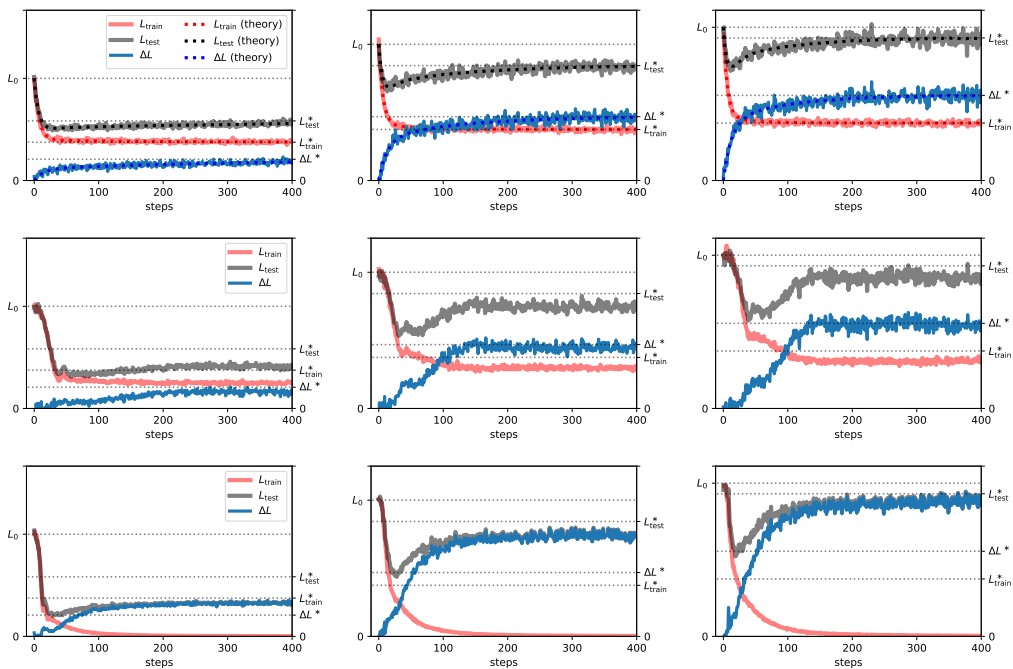

Figure 3: **ICL loss curves for different parameterizations (each row) and different $b$'s (each column).** We train (Top) the two-parameter transformer in (6), (Middle) the full-parameter transformer in (1), and (Bottom) a practical multi-head ($h = 8$) multi-layer ($\ell = 12$) transformer with softmax attention and residual connection. We use $n = 10, d = 5$, and $\sigma^2 = 0.04, 0.12, 0.16$, i.e., $b = 0.2, 0.6, 0.8$ (from Left to Right). It shows empirical results with gradient descent (solid lines) and theoretical results with gradient flow (colored dotted lines). The horizontal dotted lines indicate the four equations, $L_0 = 1 + b, L^*_{\text{train}} = \frac{d+1}{n+d+1} L_0, L^*_{\text{test}} = \frac{(d+1)L_0 + nL_r}{n+d+1}, \Delta L^* = -2b\kappa^*$. The training loss (red) and generalization loss (black) start from $L_0$ and the training loss monotonically decreases to $L^*_{\text{train}}$. On the other hand, the generalization gap (blue) monotonically increases to $\Delta L^*$. As a result, the generalization loss shows a u-shaped curve and converges to $L^*_{\text{test}}$. We use learning rate of 0.01 and batch size of 4,000. We use SGD for the two-parameter transformer, but use AdamW for the full-parameter transformer and practical models. See Figure 1 (Right) together.

*are the eigenvalues and the corresponding eigenvectors of $C_2$, respectively. To be specific, we have*

$$\alpha(t) = -2ce^{-\beta_b t} \sinh(\gamma_b t) \leq 0, \tag{11}$$

$$\kappa(t) = 2ce^{-\beta_b t} \cosh(\gamma_b t - \tau_b) + \kappa^* \geq \kappa^*, \tag{12}$$

*where $\beta_b = \frac{2+b}{2}, \gamma_b = \frac{\sqrt{4+b^2}}{2}$ and $\tau_b = \text{arctanh}\left(\frac{b}{\sqrt{4+b^2}}\right).$*

Note that the context length $n$ can only scale the dynamics through the constant $c$ and does not affect the shape of the dynamics. The proof is deferred to Appendix D.

Figure 2 shows $\alpha(t), \kappa(t)$ (Top) and the trajectory $\boldsymbol{\theta}(t)$ on the $\alpha\kappa$-plane (Bottom). Our theory (11) and (12) with gradient flow (Top, dashed) explains the empirical results with SGD (Top, solid) well.

### 4.2 Prior Forgetting and In-Context Overfitting

Theorem 4.2 illustrates the analytical dynamics of the two parameters $\alpha(t)$ and $\kappa(t)$. First, from (11), we can show that $\alpha(t)$ decreases till $t \leq t_0$ for some $t_0 > 0$ and then increases back to 0 as $t$ goes to $\infty$ since the derivative

$$\alpha'(t) = -2ce^{-\beta_b t}(-\beta_b \sinh(\gamma_b t) + \gamma_b \cosh(\gamma_b t)) = 2ce^{-\beta_b t} \sinh(\gamma_b t - \bar{\tau}_b)$$

changes its sign (from negative to positive) at $t = t_0 = \bar{\tau}_b/\gamma_b$, i.e., $\alpha'(t_0) = 0$, where $\bar{\tau}_b = \operatorname{arctanh}(\gamma_b/\beta_b) > 0$. Here the critical point $t_0$ is a decreasing function of $b$ (see Figure 5 in Appendix E), and as expected, Figure 2 (Top) shows an earlier critical point as we increase $b$.

Second, from (12), we can show that $\kappa(t)$ monotonically decreases to $\kappa^* = -n/(n + d + 1)$ since

$$\kappa'(t) = 2ce^{-\beta_b t} \left( -\beta_b \cosh(\gamma_b t - \tau_b) + \gamma_b \sinh(\gamma_b t - \tau_b) \right) = -2ce^{-\beta_b t} \cosh(\gamma_b t - \tau_b - \bar{\tau}_b) < 0.$$

Figure 2 (Top) demonstrates the above two phenomena: (i) $\alpha(t)$ decreases in the beginning and then increases back to 0 and (ii) $\kappa(t)$ monotonically decreases to $\kappa^*$. In other words, (i) the prior strength $-\alpha$ gets stronger in the beginning, but it gets weaker, reaching 0 in the later phase, which we call *prior forgetting*, and (ii) the in-context strength $-\kappa$ increases and it leads to the increase in the generalization gap (we will show this in the next section) which we call *in-context overfitting*.

### 4.3 Generalization Gap

Moreover, for the generalization gap of the model during the pretraining, from (3), (4), (5), (7), we have

$$\Delta L(\boldsymbol{\theta}) = \mathbb{E}_{\boldsymbol{w_q}, \boldsymbol{w}, \bar{X}}[\|\boldsymbol{w_q} - \hat{\boldsymbol{w}}\|^2 - \|\boldsymbol{w} - \hat{\boldsymbol{w}}\|^2] = 2\mathbb{E}_{\boldsymbol{w_q}, \boldsymbol{w}, \bar{X}}[(\boldsymbol{w_q} - \boldsymbol{w})^\top G_x(\alpha\boldsymbol{\mu} + \kappa\boldsymbol{w})] = -2b\kappa \tag{13}$$

since $\mathbb{E}[\boldsymbol{w_q}^\top \boldsymbol{w}] - \mathbb{E}[\boldsymbol{w}^\top \boldsymbol{w}] = \|\boldsymbol{\mu}\|^2 - (\|\boldsymbol{\mu}\|^2 + b) = -b$. Thus, the generalization gap increases with the in-context strength $-\kappa$.

Figure 3 (blue curves) shows the increase in the generalization gap $\Delta L$ during pretraining, which we also call *in-context overfitting*. It empirically shows that the full-parameter transformer in (1) (bottom) behaves similar to our analytical model (dotted lines) for the two-parameter transformer, also demonstrating in-context overfitting. Note that, without the two-parameter restriction, the training and generalization losses become a little smaller.

### 4.4 Different Dynamics Depending on Task Dispersion $b$

For the trajectory $\boldsymbol{\theta}(t)$ in (10), we have the derivative

$$\boldsymbol{\theta}'(t) = \underbrace{-c\lambda_+ e^{-\lambda_+ t} \boldsymbol{v}_+}_{} + \underbrace{c\lambda_- e^{-\lambda_- t} \boldsymbol{v}_-}_{}$$

with the two orthogonal components $\boldsymbol{v}_+$ and $\boldsymbol{v}_-$. At time $t$, the coefficients are $\Theta(\lambda \exp(-t\lambda))$ with respect to each $\lambda$ and we have

$$\frac{\partial}{\partial \lambda} \lambda \exp(-t\lambda) = (1 - t)\lambda \exp(-t\lambda) \begin{cases} > 0, & \text{if } t < 1, \\ < 0, & \text{if } t > 1 \end{cases}.$$

Thus, as shown in Figure 2 (Bottom), in the beginning ($t < 1$), the flow is mostly aligned with the sharper direction $-\boldsymbol{v}_+$ of the quadratic loss landscape because $\lambda_+ > \lambda_-$. On the other hand, in the later phase ($t > 1$), the opposite holds and the flow gets more aligned with the flatter one $\boldsymbol{v}_-$.

The following theorem provides intuitive pictures for the loss landscape and the gradient flow, and their changes due to variations in the value of task dispersion $b$.

**Theorem 4.3** (Evolution of Task Recognition/Learning). *The eigenvalues $\lambda_\pm$ and the corresponding eigenvectors $\boldsymbol{v}_\pm$ of the Hessian $C_2$ of the training loss $L_{train}$ are:*
*(i) for a small $b = \sigma^2 d \ll 1$,*

$$\lambda_+ = 2 + \Theta(b) \approx 2, \qquad\qquad \boldsymbol{v}_+^\top = [1, 1 + \Theta(b)] \approx [1, 1],$$
$$\lambda_- = \Theta(b) \ll \lambda_+, \qquad\qquad \boldsymbol{v}_-^\top = [1, -1 + \Theta(b)] \approx [1, -1],$$

*and, (ii) for a large $b \gg 1$,*

$$\lambda_+ = \Theta(b), \qquad\qquad \boldsymbol{v}_+^\top = [1, \Theta(b)] \parallel\sim [0, 1],$$
$$\lambda_- = 1 + \Theta(1/b) \ll \lambda_+, \qquad\qquad \boldsymbol{v}_-^\top = [1, \Theta(1/b)] \parallel\sim [1, 0],$$

*respectively, where $\boldsymbol{v} \parallel\sim \boldsymbol{u}$ means that $\boldsymbol{v}$ and $\boldsymbol{u}$ are approximately parallel.*

The proof is deferred to Appendix D.

Figure 2 (Bottom) shows that when $b$ is small, e.g., $b = 0.2$ (Left), the two main directions of the elliptic level sets are $\boldsymbol{v}_+ \approx [1,1]^\top$ and $\boldsymbol{v}_- \approx [1,-1]^\top$ and the gradient flow is first aligned with the sharper direction $\boldsymbol{v}_+$ then later it follows the flatter direction $\boldsymbol{v}_-$. And as $b$ gets larger, e.g., $b = 0.8$ (Right), the two directions becomes more like $\boldsymbol{v}_+ \parallel\sim [1,0]^\top$ and $\boldsymbol{v}_- \parallel\sim [1,0]^\top$.

Figure 2 (Top) shows that, as $b$ gets larger (from Left to Right), $\alpha(t)$ gets to have relatively smaller dip as the sharper direction is nearly aligned with $[0,1]^\top$ orthogonal to the direction increasing $\alpha$.

## 4.5 Generalization Risk

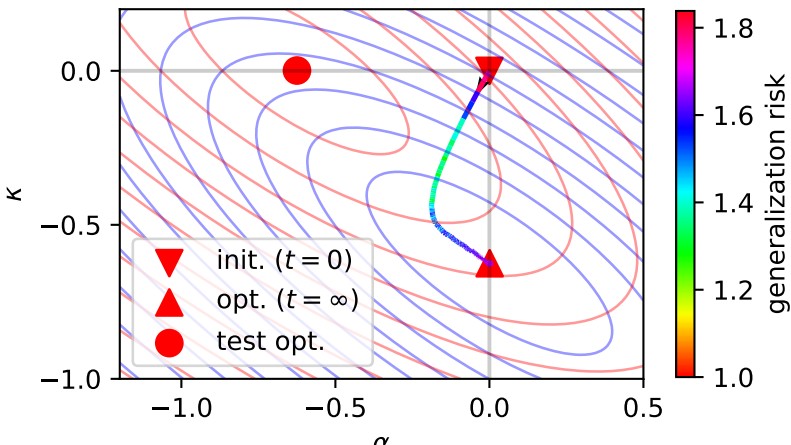

Figure 4: **Generalization risk (red ellipses) and training risk (blue ellipses).** The minimizers for the generalization risk and training risk are $[\kappa^*, 0]^\top$ ($\bullet$) and $[0, \kappa^*]^\top$ ($\blacktriangle$), respectively. The color of trajectory indicates the generalization risk value which decreases in the beginning, but increases back (in-context overfitting). See Figure 3 together.

From (13), the generalization risk $L_{\text{test}}(\boldsymbol{\theta}) = L_{\text{train}}(\boldsymbol{\theta}) - 2b\kappa$ can also be expressed as a quadratic form similar to (8):

$$L_{\text{test}}(\boldsymbol{\theta}) = \frac{1}{2}\boldsymbol{\theta}^\top C_2 \boldsymbol{\theta} + C_1'^\top \boldsymbol{\theta} + C_0 \tag{14}$$

with $C_1' = C_1 + [0, -2b]^\top = 2[1,1]^\top$ and $\arg\min_{\boldsymbol{\theta}} L_{\text{test}}(\boldsymbol{\theta}) = -C_2^{-1} C_1' = [\kappa^*, 0]^\top$.

In addition, $L_{\text{test}}^* = L_{\text{train}}^* - 2b\kappa^* = \frac{(d+1)L_0 + nL_r}{n+d+1}$ is in between $L_0$ (at zero) and $L_r$ (at random) from (9). If we are given many demonstrations with large $n$, then the trained model nearly performs the demonstration task which is a random task for the irrelevant query task.

Figure 4 visually demonstrates how the generalization risk behaves on the trajectory $\boldsymbol{\theta}(t)$. From (14), the generalization risk has the minimizer ($\bullet$) on the $\alpha$-axis, while the training risk ($\blacktriangle$) on the $\kappa$-axis. With the two elliptic level sets for the generalization risk (red) and training risk (blue), we can expect that the generalization risk decreases as $\boldsymbol{\theta}(t)$ flows along the sharper direction $-\boldsymbol{v}_+$, and then it increases as $\boldsymbol{\theta}(t)$ get close to $\boldsymbol{\theta}^*$ following $\boldsymbol{v}_-$.

## 5 Conclusion

In this paper, we investigate how the two modes of ICL emerge and disappear during pretraining. By introducing new simple settings, demonstration-query task irrelevance and noncentral task distribution, we can separately analyze the two modes of ICL and show two interesting phenomena: prior forgetting and in-context overfitting. Due to the simplicity of the analysis, we hope that our insights will motivate the future work toward understanding ICL.

## Acknowledgments

We thank the anonymous reviewers for insightful reviews. This work was partially supported by Institute of Information & communications Technology Planning & Evaluation (IITP) grants (RS-2020-II201373, Artificial Intelligence Graduate School Program (Hanyang University); RS-2023-002206284, Artificial intelligence for prediction of structure-based protein interaction reflecting physicochemical principles), the National Research Foundation of Korea (NRF) grants (RS-2023-00244896, Implicit bias of optimization algorithms for robust generalization of deep learning; the BK21 FOUR (Fostering Outstanding Universities for Research) project; NRF-2024S1A5C3A02043653, Socio-Technological Solutions for Bridging the AI Divide: A Blockchain and Federated Learning-Based AI Training Data Platform) and Korea Institute for Advanced Study (KIAS) grant funded by the Korean government (MSIT).

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

# A  Some Notations

$$d \in \mathbb{N} \qquad \text{(covariate/task dimension)}$$

$$n \in \mathbb{N} \qquad \text{(number of demonstrations)}$$

$$Z, Z_{\text{test}} \in \mathbb{R}^{(d+1) \times (n+1)} \qquad \text{(input context matrix)}$$

$$X, X_{\text{test}} \in \mathbb{R}^{d \times (n+1)}, \bar{X} \in \mathbb{R}^{d \times n}$$

$$Y, Y_{\text{test}} \in \mathbb{R}^{1 \times (n+1)}, \bar{Y} \in \mathbb{R}^{1 \times (n+1)}$$

$$\boldsymbol{x}^{(i)}, \boldsymbol{x}_{\text{test}}^{(i)} \in \mathbb{R}^d \qquad \text{(in-context covariate; } i = 1, 2, \cdots, n\text{)}$$

$$\boldsymbol{x}^{(n+1)}, \boldsymbol{x}_{\text{test}}^{(n+1)} \in \mathbb{R}^d \qquad \text{(query input)}$$

$$\mathcal{D}_{\mathcal{X}} = \mathcal{N}(\boldsymbol{0}_d, I_d) \qquad \text{(covariate distribution)}$$

$$G_x = \frac{1}{n} \bar{X} \bar{X}^{\top} = \sum_{i=1}^{n} \boldsymbol{x}^{(i)} \boldsymbol{x}^{(i)\top} \in \mathbb{R}^{d \times d} \qquad \text{((uncentered) sample covariance matrix)}$$

$$\boldsymbol{w} \in \mathbb{R}^d \qquad \text{(in-context task)}$$

$$\boldsymbol{w_q} \in \mathbb{R}^d \qquad \text{(query task)}$$

$$\hat{\boldsymbol{w}} \in \mathbb{R}^d \qquad \text{(task performed by the transformer)}$$

$$\mathcal{D}_{\mathcal{W}} = \mathcal{N}(\boldsymbol{\mu}, \Sigma_{\mathcal{W}}) \qquad \text{(task distribution)}$$

$$\boldsymbol{\mu} \in \mathbb{R}^d \qquad \text{(task center)}$$

$$\Sigma_{\mathcal{W}} = \sigma^2 I_d \qquad \text{(task covariance)}$$

$$b = \text{tr}(\Sigma_{\mathcal{W}}) = \sigma^2 d \in \mathbb{R} \qquad \text{(task dispersion)}$$

$$y^{(i)} = \boldsymbol{w}^{\top} \boldsymbol{x}^{(i)} \in \mathbb{R} \qquad \text{(target response; } i = 1, 2, \cdots, n+1\text{)}$$

$$y_{\text{test}}^{(i)} = \boldsymbol{w}^{\top} \boldsymbol{x}_{\text{test}}^{(i)} \in \mathbb{R} \qquad (i = 1, 2, \cdots, n)$$

$$y_{\text{test}}^{(n+1)} = \boldsymbol{w_q}^{\top} \boldsymbol{x}_{\text{test}}^{(n+1)} \in \mathbb{R}$$

$$T_{P,Q} : R^{(d+1) \times (n+1)} \to \mathbb{R} \qquad \text{(single-layer (full-parameter) linear Transformer)}$$

$$T_{\boldsymbol{\theta}} : R^{(d+1) \times (n+1)} \to \mathbb{R} \qquad \text{(two-parameter Transformer)}$$

$$\text{LSA}_{P,Q}(Z) = PZMZ^{\top}QZ \qquad \text{(linear self-attention)}$$

$$P = \begin{bmatrix} \boldsymbol{0}_{d \times d} & \boldsymbol{0}_d \\ \boldsymbol{p}^{\top} & \kappa \end{bmatrix} \in \mathbb{R}^{(d+1) \times (d+1)} \qquad (P \text{ for the full-parameter transformer})$$

$$P = \begin{bmatrix} \boldsymbol{0}_{d \times d} & \boldsymbol{0}_d \\ \alpha \boldsymbol{\mu}^{\top} & \kappa \end{bmatrix} \in \mathbb{R}^{(d+1) \times (d+1)} \qquad (P \text{ for the two-parameter transformer})$$

$$Q = \begin{bmatrix} \bar{Q} & \boldsymbol{0}_d \\ \boldsymbol{q}^{\top} & 0 \end{bmatrix} \in \mathbb{R}^{(d+1) \times (d+1)} \qquad (Q \text{ for the full-parameter transformer})$$

$$Q = \begin{bmatrix} I_{d \times d} & \boldsymbol{0}_d \\ \boldsymbol{0}_d^{\top} & 0 \end{bmatrix} \in \mathbb{R}^{(d+1) \times (d+1)} \qquad (Q \text{ for the two-parameter transformer})$$

$$M = \begin{bmatrix} I_n & \boldsymbol{0}_n \\ \boldsymbol{0}_n^{\top} & 0 \end{bmatrix} \in \mathbb{R}^{(n+1) \times (n+1)} \qquad \text{(masking matrix)}$$

$$\boldsymbol{\theta} = [\alpha, \kappa]^{\top} \in \mathbb{R}^2 \qquad \text{(two-parameter parameter)}$$

$$-\alpha \in \mathbb{R} \qquad \text{(prior strength)}$$

$$-\kappa \in \mathbb{R} \qquad \text{(in-context strength)}$$

$$L_{\text{train}} \in \mathbb{R} \qquad \text{(training loss)}$$

$$L_{\text{test}} \in \mathbb{R} \qquad \text{(generalization/test loss)}$$

$$\Delta L = L_{\text{test}} - L_{\text{train}} \in \mathbb{R} \qquad \text{(generalization gap)}$$

# B    Transformer Parameterization and Task $\hat{w}$

As shown below, the first $d$ rows of $P$ and the last column of $Q$ do not affect the output $T_{P,Q}(Z)$, so we put them $\mathbf{0}$. We also show that the transformer performs $\hat{w} = -(\bar{Q}^\top + qw^{(i)\top})G_x(p + \kappa w^{(i)})$.

$$Z + \frac{1}{n}\mathrm{LSA}_{P,Q}(Z) = Z + \frac{1}{n}PZMZ^\top QZ$$

$$= \begin{bmatrix} X \\ Y \end{bmatrix} + \frac{1}{n}\begin{bmatrix} \cdots & \cdots \\ p^\top & \kappa \end{bmatrix}\begin{bmatrix} X \\ Y \end{bmatrix}\begin{bmatrix} I_n & \mathbf{0}_n \\ \mathbf{0}_n^\top & 0 \end{bmatrix}\begin{bmatrix} X^\top & Y^\top \end{bmatrix}\begin{bmatrix} \bar{Q} & \cdots \\ q^\top & \cdots \end{bmatrix}\begin{bmatrix} X \\ Y \end{bmatrix}$$

$$= \begin{bmatrix} X \\ Y \end{bmatrix} + \frac{1}{n}\begin{bmatrix} \cdots & \cdots \\ p^\top\bar{X} + \kappa\bar{Y} & p^\top x^{(n+1)} \end{bmatrix}\begin{bmatrix} I_n & \mathbf{0}_n \\ \mathbf{0}_n^\top & 0 \end{bmatrix}\begin{bmatrix} X^\top & Y^\top \end{bmatrix}\begin{bmatrix} \bar{Q} & \cdots \\ q^\top & \cdots \end{bmatrix}\begin{bmatrix} X \\ Y \end{bmatrix}$$

$$= \begin{bmatrix} X \\ Y \end{bmatrix} + \frac{1}{n}\begin{bmatrix} \cdots & \mathbf{0}_d \\ p^\top\bar{X} + \kappa\bar{Y} & 0 \end{bmatrix}\begin{bmatrix} \bar{X}^\top & \bar{Y}^\top \\ x^{(n+1)\top} & 0 \end{bmatrix}\begin{bmatrix} \bar{Q} & \cdots \\ q^\top & \cdots \end{bmatrix}\begin{bmatrix} X \\ Y \end{bmatrix}$$

$$T_{P,Q}(Z) = -\left[Z + \frac{1}{n}\mathrm{LSA}_{P,Q}(Z)\right]_{d+1,n+1}$$

$$= -Ye_{n+1} - \frac{1}{n}\begin{bmatrix} p^\top\bar{X} + \kappa\bar{Y} & 0 \end{bmatrix}\begin{bmatrix} \bar{X}^\top & \bar{Y}^\top \\ x^{(n+1)\top} & 0 \end{bmatrix}\begin{bmatrix} \bar{Q} & \cdots \\ q^\top & \cdots \end{bmatrix}\begin{bmatrix} X \\ Y \end{bmatrix}e_{n+1}$$

$$= -\frac{1}{n}\begin{bmatrix} p^\top\bar{X} + \kappa\bar{Y} & 0 \end{bmatrix}\begin{bmatrix} \bar{X}^\top & \bar{Y}^\top \\ x^{(n+1)\top} & 0 \end{bmatrix}\begin{bmatrix} \bar{Q} & \cdots \\ q^\top & \cdots \end{bmatrix}\begin{bmatrix} x^{(n+1)} \\ 0 \end{bmatrix}$$

$$= -\frac{1}{n}\begin{bmatrix} p^\top\bar{X} + \kappa\bar{Y} & 0 \end{bmatrix}\begin{bmatrix} \bar{X}^\top\bar{Q} + \bar{Y}^\top q^\top & \cdots \\ x^{(n+1)\top}\bar{Q} & \cdots \end{bmatrix}\begin{bmatrix} x^{(n+1)} \\ 0 \end{bmatrix}$$

$$= -\frac{1}{n}x^{(n+1)\top}\left[(\bar{Q}^\top\bar{X} + q\bar{Y})(\bar{X}^\top p + \kappa\bar{Y}^\top)\right]$$

$$= -\frac{1}{n}x^{(n+1)\top}\left[\bar{Q}^\top\bar{X}\bar{X}^\top p + \kappa\bar{Q}^\top\bar{X}\bar{Y}^\top + q\bar{Y}\bar{X}^\top p + \kappa q\bar{Y}\bar{Y}^\top\right]$$

$$= x^{(n+1)\top}\left[-\frac{1}{n}\sum_{i=1}^{n}\bar{Q}^\top\bar{x}^{(i)}x^{(i)\top}p + \kappa y^{(i)}\bar{Q}^\top x^{(i)} + y^{(i)}qx^{(i)\top}p + \kappa qy^{(i)2}\right]$$

$$= x^{(n+1)\top}\left[-\frac{1}{n}\sum_{i=1}^{n}\bar{Q}^\top\bar{x}^{(i)}x^{(i)\top}p + \kappa w^\top x^{(i)}\bar{Q}^\top x^{(i)} + w^\top x^{(i)}qx^{(i)\top}p + \kappa qw^\top x^{(i)}x^{(i)\top}w\right]$$

$$= x^{(n+1)\top}\left[-\frac{1}{n}\sum_{i=1}^{n}(\bar{Q}^\top\bar{x}^{(i)} + qw^\top x^{(i)})(x^{(i)\top}p + \kappa x^{(i)\top}w)\right]$$

$$= x^{(n+1)\top}\left[-\frac{1}{n}\sum_{i=1}^{n}(\bar{Q}^\top + qw^\top)x^{(i)}x^{(i)\top}(p + \kappa w)\right]$$

$$= x^{(n+1)\top}\underbrace{\left[-(\bar{Q}^\top + qw^\top)G_x(p + \kappa w)\right]}_{\hat{w}}$$

which proves the equation (2). Moreover, for the two-parameter transformer with $\bar{Q} = I_d, q = \mathbf{0}_d, p = \alpha\mu$, we have

$$\hat{w} = -(\bar{Q}^\top + qw^\top)G_x(p + \kappa w)$$
$$= -G_x(\alpha\mu + \kappa w)$$

which proves the equation (7).

# C   Higher Moments of Multivariate Gaussian

**Lemma C.1.** *For $\boldsymbol{w} \sim \mathcal{N}(\boldsymbol{\mu}, \sigma^2 I_d)$ and a matrix $A \in \mathbb{R}^{d \times d}$, we have the first four moments as follows:*

$$\mathbb{E}[\boldsymbol{w}] = \boldsymbol{\mu},$$
$$\mathbb{E}[\boldsymbol{w}\boldsymbol{w}^\top] = \boldsymbol{\mu}\boldsymbol{\mu}^\top + \sigma^2 I_d =: M_\sigma,$$
$$\mathbb{E}[\|\boldsymbol{w}\|^2] = \|\boldsymbol{\mu}\|^2 + d\sigma^2,$$
$$\mathbb{E}[\boldsymbol{w}^\top A\boldsymbol{w}] = \boldsymbol{\mu}^\top A\boldsymbol{\mu} + \sigma^2 \operatorname{Tr}(A),$$
$$\mathbb{E}[\boldsymbol{w}\boldsymbol{w}^\top \boldsymbol{w}] = (2\sigma^2 + d\sigma^2 + \|\boldsymbol{\mu}\|^2)\boldsymbol{\mu},$$
$$\mathbb{E}[\boldsymbol{w}\boldsymbol{w}^\top A\boldsymbol{w}] = \sigma^2(A + A^\top)\boldsymbol{\mu} + (\sigma^2 \operatorname{Tr}(A) + \boldsymbol{\mu}^\top A\boldsymbol{\mu})\boldsymbol{\mu},$$
$$\mathbb{E}[\boldsymbol{w}\boldsymbol{w}^\top \boldsymbol{w}\boldsymbol{w}^\top] = 2M_\sigma^2 - 2m^2 + (d\sigma^2 + \|\boldsymbol{\mu}\|^2)M_\sigma,$$
$$\mathbb{E}[\boldsymbol{w}^\top \boldsymbol{w}\boldsymbol{w}^\top \boldsymbol{w}] = 2\sigma^2(2\|\boldsymbol{\mu}\|^2 + d\sigma^2) + (d\sigma^2 + \|\boldsymbol{\mu}\|^2)^2,$$
$$\mathbb{E}[\boldsymbol{w}\boldsymbol{w}^\top A\boldsymbol{w}\boldsymbol{w}^\top] = M_\sigma(A + A^\top)M_\sigma - m(A + A^\top)m + \operatorname{Tr}(M_\sigma A)M_\sigma,$$

*where $m = \boldsymbol{\mu}\boldsymbol{\mu}^\top$.*

*If $\sigma^2 = 1$, then we have*

$$\mathbb{E}[\boldsymbol{w}] = \boldsymbol{\mu},$$
$$\mathbb{E}[\boldsymbol{w}\boldsymbol{w}^\top] = \boldsymbol{\mu}\boldsymbol{\mu}^\top + I_d =: M,$$
$$\mathbb{E}[\|\boldsymbol{w}\|^2] = \|\boldsymbol{\mu}\|^2 + d,$$
$$\mathbb{E}[\boldsymbol{w}^\top A\boldsymbol{w}] = \boldsymbol{\mu}^\top A\boldsymbol{\mu} + \operatorname{Tr}(A),$$
$$\mathbb{E}[\boldsymbol{w}\boldsymbol{w}^\top \boldsymbol{w}] = (2 + d + \|\boldsymbol{\mu}\|^2)\boldsymbol{\mu},$$
$$\mathbb{E}[\boldsymbol{w}\boldsymbol{w}^\top A\boldsymbol{w}] = (A + A^\top)\boldsymbol{\mu} + (\operatorname{Tr}(A) + \boldsymbol{\mu}^\top A\boldsymbol{\mu})\boldsymbol{\mu},$$
$$\mathbb{E}[\boldsymbol{w}\boldsymbol{w}^\top \boldsymbol{w}\boldsymbol{w}^\top] = 2M^2 - 2m^2 + (d + \|\boldsymbol{\mu}\|^2)M,$$
$$\mathbb{E}[\boldsymbol{w}^\top \boldsymbol{w}\boldsymbol{w}^\top \boldsymbol{w}] = 2(2\|\boldsymbol{\mu}\|^2 + d) + (d + \|\boldsymbol{\mu}\|^2)^2,$$
$$\mathbb{E}[\boldsymbol{w}\boldsymbol{w}^\top A\boldsymbol{w}\boldsymbol{w}^\top] = M(A + A^\top)M - m(A + A^\top)m + \operatorname{Tr}(MA)M.$$

*If we further assume $\boldsymbol{\mu} = \mathbf{0}_d$, then we have*

$$\mathbb{E}[\boldsymbol{w}] = \mathbf{0}_d,$$
$$\mathbb{E}[\boldsymbol{w}\boldsymbol{w}^\top] = I_d,$$
$$\mathbb{E}[\|\boldsymbol{w}\|^2] = d,$$
$$\mathbb{E}[\boldsymbol{w}^\top A\boldsymbol{w}] = \operatorname{Tr}(A),$$
$$\mathbb{E}[\boldsymbol{w}\boldsymbol{w}^\top \boldsymbol{w}] = \mathbf{0}_d,$$
$$\mathbb{E}[\boldsymbol{w}\boldsymbol{w}^\top A\boldsymbol{w}] = \mathbf{0}_d,$$
$$\mathbb{E}[\boldsymbol{w}\boldsymbol{w}^\top \boldsymbol{w}\boldsymbol{w}^\top] = (2 + d)I_d,$$
$$\mathbb{E}[\boldsymbol{w}^\top \boldsymbol{w}\boldsymbol{w}^\top \boldsymbol{w}] = 2d + d^2,$$
$$\mathbb{E}[\boldsymbol{w}\boldsymbol{w}^\top A\boldsymbol{w}\boldsymbol{w}^\top] = A + A^\top + \operatorname{Tr}(A)I_d.$$

## D    Proofs

*Proof of Theorem 4.1.* From (4), we have

$$
\begin{aligned}
L_{\text{train}}(P, Q) &= \mathbb{E}_{\boldsymbol{w}, \bar{X}}[\|\boldsymbol{w} - \hat{\boldsymbol{w}}\|^2] \\
&= \mathbb{E}_{\boldsymbol{w}, \bar{X}}[\|\boldsymbol{w}\|^2 + \|\hat{\boldsymbol{w}}\|^2 - 2\boldsymbol{w}^\top \hat{\boldsymbol{w}}] \\
&= \underbrace{\mathbb{E}_{\boldsymbol{w}, \bar{X}}[\|\hat{\boldsymbol{w}}\|^2]}_{\text{1st}} \underbrace{- 2\mathbb{E}_{\boldsymbol{w}, \bar{X}}[\boldsymbol{w}^\top \hat{\boldsymbol{w}}]}_{\text{2nd}} + \underbrace{\mathbb{E}_{\boldsymbol{w}}[\|\boldsymbol{w}\|^2]}_{\text{3rd}}.
\end{aligned}
$$

From (7), we can get the first term as follows:

$$
\begin{aligned}
\hat{\boldsymbol{w}} &= -G_x(\alpha\boldsymbol{\mu} + \kappa\boldsymbol{w}), \\
\|\hat{\boldsymbol{w}}\|^2 &= (\alpha\boldsymbol{\mu} + \kappa\boldsymbol{w})^\top G_x^2 (\alpha\boldsymbol{\mu} + \kappa\boldsymbol{w}) \\
&= \kappa^2 \boldsymbol{w}^\top G_x^2 w + 2\kappa\alpha\boldsymbol{\mu}^\top G_x^2 \boldsymbol{w} + \alpha^2 \boldsymbol{\mu}^\top G_x^2 \boldsymbol{\mu}, \\
\mathbb{E}_{\bar{X}}[\|\hat{\boldsymbol{w}}\|^2] &= \kappa^2 \boldsymbol{w}^\top \mathbb{E}_{\bar{X}}[G_x^2]\boldsymbol{w} + 2\kappa\alpha\boldsymbol{\mu}^\top \mathbb{E}_{\bar{X}}[G_x^2]\boldsymbol{w} + \alpha^2 \boldsymbol{\mu}^\top \mathbb{E}_{\bar{X}}[G_x^2]\boldsymbol{\mu}, \\
\mathbb{E}_{\boldsymbol{w}, \bar{X}}[\|\hat{\boldsymbol{w}}\|^2] &= \kappa^2 (G_\mu + \text{tr}(G)\sigma^2) + 2\kappa\alpha G_\mu + \alpha^2 G_\mu && \text{(Lemma C.1)} \\
&= (\kappa + \alpha)^2 G_\mu + \kappa^2 \text{tr}(G)\sigma^2 \\
&= \left((\kappa + \alpha)^2 + \kappa^2 \sigma^2 d\right) \frac{n + d + 1}{n} \\
&= \frac{n + d + 1}{n} \begin{bmatrix} \alpha & \kappa \end{bmatrix} \begin{bmatrix} 1 & 1 \\ 1 & 1 + b \end{bmatrix} \begin{bmatrix} \alpha \\ \kappa \end{bmatrix} \\
&= \frac{1}{2}\boldsymbol{\theta}^\top C_2 \boldsymbol{\theta},
\end{aligned}
$$

where

$$
\begin{aligned}
G = \mathbb{E}_{\bar{X}}[G_x^2] &= \frac{1}{n}\mathbb{E}_{\bar{X}}[\boldsymbol{x}\boldsymbol{x}^\top \boldsymbol{x}\boldsymbol{x}^\top + (n-1)\boldsymbol{x}\boldsymbol{x}^\top \boldsymbol{x}'\boldsymbol{x}'^\top] \\
&= \frac{1}{n}[(d+2)I_d + (n-1)I_d] && \text{(Lemma C.1)} \\
&= \frac{n + d + 1}{n} I_d, \\
\text{tr}(G) &= \frac{n + d + 1}{n} d, \\
G_\mu \equiv \boldsymbol{\mu}^\top G\boldsymbol{\mu} &= \frac{n + d + 1}{n}.
\end{aligned}
$$

The second term is

$$
\begin{aligned}
-\boldsymbol{w}^\top \hat{\boldsymbol{w}} &= \boldsymbol{w}^\top G_x(\boldsymbol{p} + \kappa\boldsymbol{w}) \\
&= \boldsymbol{w}^\top G_x \boldsymbol{p} + \kappa\boldsymbol{w}^\top G_x \boldsymbol{w} \\
&= \alpha\boldsymbol{w}^\top G_x \boldsymbol{\mu} + \kappa\boldsymbol{w}^\top G_x \boldsymbol{w}, \\
-\mathbb{E}_{\bar{X}}[\boldsymbol{w}^\top \hat{\boldsymbol{w}}] &= \alpha\boldsymbol{w}^\top \mathbb{E}[G_x]\boldsymbol{\mu} + \kappa\boldsymbol{w}^\top \mathbb{E}[G_x]\boldsymbol{w} \\
&= \alpha\boldsymbol{w}^\top \boldsymbol{\mu} + \kappa\boldsymbol{w}^\top \boldsymbol{w}, \\
-\mathbb{E}_{\boldsymbol{w}, \bar{X}}[\boldsymbol{w}^\top \hat{\boldsymbol{w}}] &= \alpha + \kappa(1 + b) \\
&= \begin{bmatrix} 1 & 1 + b \end{bmatrix} \begin{bmatrix} \alpha \\ \kappa \end{bmatrix} \\
&= \frac{1}{2}C_1^\top \boldsymbol{\theta}.
\end{aligned}
$$

The last term is $\mathbb{E}_{\boldsymbol{w}}[\|\boldsymbol{w}\|^2] = 1 + b = C_0$.

Next, we want to calculate the minimizer $\boldsymbol{\theta}^*$ and the corresponding minimum value. First, as $C_2 \succ 0$, the training loss has only one critical point which is the minimizer:

$$\nabla_{\boldsymbol{\theta}} L_{\text{train}}(\boldsymbol{\theta}^*) = C_2 \boldsymbol{\theta}^* + C_1 = 0,$$

$$\boldsymbol{\theta}^* = -C_2^{-1} C_1$$

$$= -\frac{n}{n+d+1} \begin{bmatrix} 1 & 1 \\ 1 & 1+b \end{bmatrix}^{-1} \begin{bmatrix} 1 \\ 1+b \end{bmatrix}$$

$$= -\frac{n}{n+d+1} \frac{1}{b} \begin{bmatrix} 1+b & -1 \\ -1 & 1 \end{bmatrix} \begin{bmatrix} 1 \\ 1+b \end{bmatrix}$$

$$= -\frac{n}{n+d+1} \frac{1}{b} \begin{bmatrix} 0 \\ b \end{bmatrix}$$

$$= -\frac{n}{n+d+1} \begin{bmatrix} 0 \\ 1 \end{bmatrix}.$$

We can calculate the corresponding minimum value at the minimizer by plugging in $\boldsymbol{\theta}^* = -C^{-2} C_1$ and $\boldsymbol{\theta}^* = -[0, -n/(n+d+1)]^\top$ into $L_{\text{train}}(\boldsymbol{\theta}^*)$ as follows:

$$L_{\text{train}}(\boldsymbol{\theta}^*) = \frac{1}{2} \boldsymbol{\theta}^{*\top} C_2 \boldsymbol{\theta}^* + C_1^\top \boldsymbol{\theta}^* + C_0$$

$$= -\frac{1}{2} C_1^\top C_2^{-1} C_1 + C_0$$

$$= \frac{1}{2} C_1^\top \boldsymbol{\theta}^* + 1 + b$$

$$= -(1+b) \frac{n}{n+d+1} + 1 + b$$

$$= \frac{(1+b)(d+1)}{n+d+1}$$

$$= \frac{d+1}{n+d+1} L_0.$$

$\square$

*Proof of Theorem 4.2.*

$$\dot{\boldsymbol{\theta}} = -\nabla_{\boldsymbol{\theta}} L_{\text{train}}(\boldsymbol{\theta}) = -C_2 \boldsymbol{\theta} - C_1,$$

$$\boldsymbol{\theta}(t) = c_1 e^{\lambda_1 t} \boldsymbol{v}_1 + c_2 e^{\lambda_2 t} \boldsymbol{v}_2 + \boldsymbol{\theta}^*$$

$$= c_1 e^{\lambda_1 t} \begin{bmatrix} 1 \\ \frac{b+\sqrt{4+b^2}}{2} \end{bmatrix} + c_2 e^{\lambda_2 t} \begin{bmatrix} 1 \\ \frac{b-\sqrt{4+b^2}}{2} \end{bmatrix} + \begin{bmatrix} 0 \\ -\frac{n}{n+d+1} \end{bmatrix},$$

where $\lambda_1, \lambda_2$ are eigenvalues of $-C_2$, i.e., $\lambda_1 = -\lambda_+, \lambda_2 = -\lambda_-$, and $c_1, c_2$ are determined by the initial condition

$$\boldsymbol{\theta}(0) = c_1 \begin{bmatrix} 1 \\ \frac{b+\sqrt{4+b^2}}{2} \end{bmatrix} + c_2 \begin{bmatrix} 1 \\ \frac{b-\sqrt{4+b^2}}{2} \end{bmatrix} + \begin{bmatrix} 0 \\ -\frac{n}{n+d+1} \end{bmatrix} = \begin{bmatrix} 0 \\ 0 \end{bmatrix}.$$

Therefore, solving the following system of linear equations

$$\begin{cases} c_1 + c_2 & = 0 \\ c_1 \frac{b+\sqrt{4+b^2}}{2} + c_2 \frac{b-\sqrt{4+b^2}}{2} - \frac{n}{n+d+1} & = 0 \end{cases},$$

we have

$$c_1 = -c_2 = \frac{n}{(n+d+1)\sqrt{4+b^2}}.$$

Therefore, we have

$$\boldsymbol{\theta}(t) = \frac{n}{n+d+1}\left(\frac{1}{\sqrt{4+b^2}}e^{\lambda_1 t}\begin{bmatrix}1\\\frac{b+\sqrt{4+b^2}}{2}\end{bmatrix} - \frac{1}{\sqrt{4+b^2}}e^{\lambda_2 t}\begin{bmatrix}1\\\frac{b-\sqrt{4+b^2}}{2}\end{bmatrix} + \begin{bmatrix}0\\-1\end{bmatrix}\right),$$

$$\alpha(t) = \frac{n}{n+d+1}\frac{1}{\sqrt{4+b^2}}e^{-\frac{2+b}{2}t}\left(e^{-\frac{\sqrt{4+b^2}}{2}t} - e^{\frac{\sqrt{4+b^2}}{2}t}\right)$$

$$= -\frac{n}{n+d+1}\frac{2}{\sqrt{4+b^2}}e^{-\frac{2+b}{2}t}\sinh\left(\frac{\sqrt{4+b^2}}{2}t\right),$$

$$\kappa(t) = \frac{n}{n+d+1}\frac{1}{\sqrt{4+b^2}}\left(\frac{b}{2}e^{-\frac{2+b}{2}t}\left(e^{-\frac{\sqrt{4+b^2}}{2}t} - e^{\frac{\sqrt{4+b^2}}{2}t}\right) + \frac{\sqrt{4+b^2}}{2}e^{-\frac{2+b}{2}t}\left(e^{-\frac{\sqrt{4+b^2}}{2}t} + e^{\frac{\sqrt{4+b^2}}{2}t}\right)\right) + \kappa^*$$

$$= \frac{n}{n+d+1}\frac{2}{\sqrt{4+b^2}}e^{-\frac{2+b}{2}t}\left(-\frac{b}{2}\sinh\left(\frac{\sqrt{4+b^2}}{2}t\right) + \frac{\sqrt{4+b^2}}{2}\cosh\left(\frac{\sqrt{4+b^2}}{2}t\right)\right) + \kappa^*$$

$$= \frac{n}{n+d+1}\frac{2}{\sqrt{4+b^2}}e^{-\frac{2+b}{2}t}\left(-\sinh(\tau_b)\sinh\left(\frac{\sqrt{4+b^2}}{2}t\right) + \cosh(\tau_b)\cosh\left(\frac{\sqrt{4+b^2}}{2}t\right)\right) + \kappa^*$$

$$= \frac{n}{n+d+1}\frac{2}{\sqrt{4+b^2}}e^{-\frac{2+b}{2}t}\cosh\left(\frac{\sqrt{4+b^2}}{2}t - \tau_b\right) + \kappa^*.$$

$\square$

*Proof of Theorem 4.3.* If $b$ is small, then we can ignore higher order terms and obtain

$$(4+b^2)^{1/2} = 2(1+(b/2)^2)^{1/2} = 2 + \Theta(b^2)$$

and $\frac{b\pm\sqrt{4+b^2}}{2} = \pm 1 + \Theta(b)$. For large $b$, we have

$$(4+b^2)^{1/2} = b(4b^{-2}+1)^{1/2} = b + \Theta(b^{-1})$$

and thus, $b+\sqrt{4+b^2} = \Theta(b)$ and $b-\sqrt{4+b^2} = \Theta(b^{-1})$. To summarize, we have the eigenvectors

$$\boldsymbol{v}_+ = \begin{bmatrix}1\\\frac{b+\sqrt{4+b^2}}{2}\end{bmatrix} = \begin{cases}[1,\ 1+\Theta(b)]^\top & \text{if } b \ll 1\\ [1,\ \Theta(b)]^\top & \text{if } b \gg 1\end{cases},$$

$$\boldsymbol{v}_- = \begin{bmatrix}1\\\frac{b-\sqrt{4+b^2}}{2}\end{bmatrix} = \begin{cases}[1,\ -1+\Theta(b)]^\top & \text{if } b \ll 1\\ [1,\ \Theta(1/b)]^\top & \text{if } b \gg 1\end{cases}.$$

Similarly, for the eigenvalues, we have $\lambda_\pm = \frac{2+b\pm\sqrt{4+b^2}}{2} = 1 + \frac{b\pm\sqrt{4+b^2}}{2}$, and thus

$$\lambda_+ = \begin{cases}2+\Theta(b) & \text{if } b \ll 1\\ \Theta(b) & \text{if } b \gg 1\end{cases},$$

$$\lambda_- = \begin{cases}\Theta(b) & \text{if } b \ll 1\\ 1+\Theta(1/b) & \text{if } b \gg 1\end{cases}.$$

$\square$

## E  Extra Figures

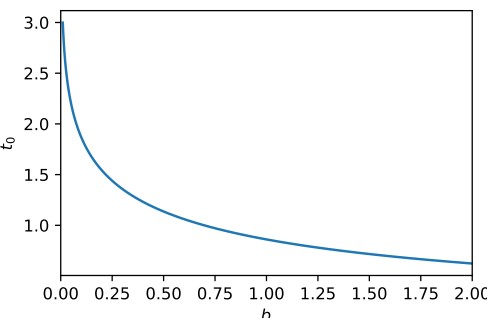

Figure 5: The critical point $t_0 = \frac{1}{\gamma_b}\mathrm{arctanh}(\gamma_b/\beta_b) = \frac{2}{\sqrt{4+b^2}}\mathrm{arctanh}\left(\frac{\sqrt{4+b^2}}{2+b}\right)$ that $\alpha'(t)$ changes its sign from negative to positive becomes earlier as we increase the value of $b$.

