# OpenReview forum: "Prior Forgetting and In-Context Overfitting"
_NeurIPS.cc/2025/Conference — NeurIPS 2025 poster_

### Official Review · Reviewer_LaxR · 2025-06-10

**Clarity:** 3
**Significance:** 2
**Originality:** 2
**Rating:** 5
**Confidence:** 4

**Summary:**

This paper investigates two modes of in-context learning (ICL) exhibited by pre-trained Transformer models during testing, with a particular focus on their emergence and dynamic evolution during the pre-training phase.
The study first analyzes a simple random linear regression problem using a linear attention mechanism, and then extends to more complex function classes such as two-layer neural networks, decision trees, exponential regression, softmax regression, and autoregressive learning.
In addition, the authors introduce a new setting called "demonstration-query task irrelevance" to examine the model's robustness and generalization under various distribution shifts.
They emphasize the balance of power between the two ICL modes and propose a measurement method to evaluate their strength and how they emerge or fade over time.
Through theoretical analysis and experimental validation, the authors demonstrate the phenomenon known as "in-context overfitting", where the model's generalization gap increases with greater context strength.

**Questions:**

See Weaknesses


If the authors could **provide experiments on larger-scale LLMs and more complex datasets**, I would be willing to raise my score.

**Ethical Concerns:**

["NO or VERY MINOR ethics concerns only"]

**Final Justification:**

My concerns are fully addressed.

**Limitations:**

Yes

**Quality:**

3

**Strengths And Weaknesses:**

**Strengths**

+ The paper provides a deep theoretical understanding of the two in-context learning (ICL) modes in pre-trained Transformers, supported by rigorous empirical validation across a variety of task distributions and function classes.

+ The authors introduce a new evaluation setting called "demonstration-query task irrelevance", which enables the study of model robustness and generalization under distributional shifts.

**Weaknesses**

+ While the paper provides a detailed theoretical analysis of the two-parameter Transformer model and demonstrates similar behavior to the full-parameter model through experiments, **this simplification may overlook some dynamic changes and characteristics present in more complex models**. Consequently, the findings might not fully represent the behaviors of the more intricate Transformer architectures used in practical applications. Additionally, it's worth noting that LLMs' pre-training typically involves training on datasets at the billion-token scale. Although this study claims to investigate phenomena during the pre-training phase, **its setup does not entirely align with conventional LLM pre-training practices**, which could be considered one of its primary limitations.

+ It would be beneficial for the authors to employ **more complex datasets** to analyze the dynamics of in-context learning and evaluate Task Recognition and Task Learning using setups similar to those in other relevant studies [1, 2].

+ Although the paper mentions several hyperparameters, such as prior strength and in-context strength, there could be more extensive experimentation to explore how these parameters influence model performance.

[1] What In-Context Learning "Learns" In-Context: Disentangling Task Recognition and Task Learning, ACL 2023.

[2] Investigating the Pre-Training Dynamics of In-Context Learning: Task Recognition vs. Task Learning, ICLR 2025.

---

> ### Author Rebuttal · Authors · 2025-07-28
>
> > While the paper provides a detailed theoretical analysis of the two-parameter Transformer model and demonstrates similar behavior to the full-parameter model through experiments, this simplification may overlook some dynamic changes and characteristics present in more complex models. Consequently, the findings might not fully represent the behaviors of the more intricate Transformer architectures used in practical applications. Additionally, it's worth noting that LLMs' pre-training typically involves training on datasets at the billion-token scale. Although this study claims to investigate phenomena during the pre-training phase, its setup does not entirely align with conventional LLM pre-training practices, which could be considered one of its primary limitations.
>
> > It would be beneficial for the authors to employ more complex datasets to analyze the dynamics of in-context learning and evaluate Task Recognition and Task Learning using setups similar to those in other relevant studies [1, 2].
>
> [A1+A2] Thank you for the feedback. Relaxing the simple setting, we trained a more complex and practical **multi-head ($h=8$) multi-layer ($\ell\geq12$) Transformer with softmax activation and residual connection with larger $n\geq 256$ and about a billion training tokens ($\geq$ 1B)** (we used AdamW to accelerate the optimization), and **observed in-context overfitting phenomenon** similar to what is shown in Figure 3 ($L_{\text{test}}$ decreases and then increases). We will include a discussion for more complex models (including the above) in the revised version.
>
> Due to limited time and resources, we were unable to analyze more complex datasets, but it is also expected to observe transient nature of in-context learning for more complex tasks other than the simple regression task as similar phenomena have already been demonstrated empirically in previous studies (e.g, [a] for a synthetic Omniglot classification tasks and [b] for a Markov mixture tasks).
>
> [a] Singh et al., The transient nature of emergent in-context learning in transformers (NeurIPS 2023)
>
> [b] Park et al., Competition dynamics shape algorithmic phases of in-context learning (ICLR 2025)
>
>
> ---
> > Although the paper mentions several hyperparameters, such as prior strength and in-context strength, there could be more extensive experimentation to explore how these parameters influence model performance.
>
> [A3] First of all, prior strength $|\alpha|$ and in-context strength $|\kappa|$ are **not hyperparameters** that we set before training. They are **parameters** that the model learns during training.
>
> - (How $\kappa$ influence model performance) For example, the generalization gap $\Delta L = -2b \kappa=2b|\kappa|$ is proportional to $|\kappa|$ so the generalization gap increases with the in-context strength $|\kappa|$ (in-context overfitting).
>
> - (Hyperparameters 1)
> Theoretically, we have the following four hyperparameters: $\sigma^2,d,n$, and $b=\sigma^2 d$.
> Theorem 4.1-4.2 and Figure 2-3 show that the task dispersion $b=\sigma^2 d$ (controlled jointly by task variance $\sigma^2$ and task dimension $d$) plays an important role in determining the objective $L_{\text{train}}(\theta)$ and in shaping the trajectory $\theta(t)$.
> Eq (10) can be reformulated as follows:
> $$\theta(t)=c*\underbracket{ (e^{-\lambda\_+t}v\_+-e^{-\lambda\_-t}v\_-+[0,\sqrt{4+b^2}]^\top) }\_{\text{only depends on $b$, not on $n$}}=c*f(t;b),\ \text{where the scaling factor}\ c=\frac{n}{n+d+1}\frac{1}{\sqrt{4+b^2}}$$
> since $\lambda_{\pm}$ and $v\_\pm$ only depend on $b$, not $n$.
> Therefore, the task dispersion $b=\sigma^2 d$ ($\sigma^2$ and $d$ together) **control the shape of the trajectory $\theta(t)$**, but the context length $n$ can **only scale the dynamics** and does not affect the shape of the dynamics.
> Even in the case of $n\gg d$, the scaling factor $c\approx \frac{1}{\sqrt{4+b^2}}$ shows minimal sensitivity to $n$, and $n$ has little to no effect to control the scale.
> | Hyperparam. | $b= \mathrm{tr}(\Sigma_{\mathcal{W}})=\sigma^2d$ | $n$ |
> | --- | --- | --- |
> | Name | task dispersion = (task variance)*(task dimension) | number of demonstration (context length) |
> | Control | shape of the trajectory | scale of the trajectory |
>
> - (Hyperparameters 2) Empirically, we can name the following additional hyperparameters and **summarize their influence on the dynamics**:
> batch size, learning rate, optimizers (and their internal hyperparameters), initialization, number of layers, non-linear activation (e.g., softmax).
>     - [batch size and learning rate] As our theory is based on the gradient flow, when we used smaller batch size or larger learning rate, we observed more noisy curves for Figure 3, but they, if smoothed, still follow the theoretical curve and the fundamental understanding remains the same.
>     - [optimizer] As our theory is based on the gradient flow, when we used adaptive methods (e.g., Adam or AdamW), we observed loss curves somewhat different from the one in Fig 3, but still observed in-context overfitting with similar values of $L_{\text{train}}^\ast,L_{\text{test}}^\ast,\Delta L^\ast$.
>     - [initialization] not very sensitive and it just needs to be small
>     - [number of layers] Using multi-layer Transformer, we observed a loss plateau (before a sudden loss drop). This plateau is predicted by the theory of layerwise linear neural network [c,d]. Especially, [c] shows that single-layer linear transformer is equivalent to the two-layer linear neural network (LNN) with cubic feature input. Let's consider the simplest setup: we want to minimize $L(a,b)=(1-bax)^2$ where $a,b$ represent the parameters of each layer in the two-layer LNN, $x$ the input, and 1 the target ($x\mapsto ax\mapsto bax$). Then, $\dot a =-\frac{\partial L}{\partial a}=2(1-abx)bx$ and $\dot b=-\frac{\partial L}{\partial b}=2(1-abx)ax$. Thus, $c=ab$ follows the dynamics $\dot c = a\dot b+ b\dot a = 4(1-cx)cx$ which has a sigmoidal solution of $c(t)=\frac{1}{x(C_1\exp(-4xt)+1)}$. Thus, the loss starts with a (long) plateau, exhibits a sudden drop, and saturates. This intuition also holds for general LNNs. Note that we didn't use SGD to train the full-parameter transformer in Figure 3. We will add the corresponding plots that showed a plateau (by using SGD to train the full-parameter Transformer), together with the above discussion.
>     - [non-linear activation (softmax-based attention)] Empirical results for softmax-based attention also show in-context overfitting qualitatively similar with the theoretical prediction for the two-parameter model (with similar values of $L_{\text{train}}^\ast,L_{\text{test}}^\ast,\Delta L^\ast$).
>
> [c] Saxe et al., Exact solutions to the nonlinear dynamics of learning in deep linear neural networks (ICLR 2014)
>
> [d] Zhang et al., Training Dynamics of In-Context Learning in Linear Attention (ICML 2025)

---

> > ### Comment · Reviewer_LaxR · 2025-08-02
> > **Official Comment**
> >
> > My concerns are fully addressed. Thus, I will consider uprating my score.

---

### Official Review · Reviewer_25wX · 2025-06-21

**Clarity:** 3
**Significance:** 2
**Originality:** 3
**Rating:** 4
**Confidence:** 3

**Summary:**

This theoretical paper considers in-context learning (ICL) and argues, using a simple theoretical model, that two distinct modes of ICL dominate different stages of pretraining. In the early stage, the model relies more on task identification than on in-context examples. In the later stage, in-context examples become more important. These two ICL modes have been previously observed and described. The main novelty is in developing a simple model that captures these behaviors and how they emerge and dominate different pretraining stages. The theory is validated with some experiments.

**Questions:**

- In Fig 3, the generalization gaps for two and full-parameter attentions are very similar. Is this a coincidence, or are there reasons to expect this result?
- The demonstration-query task irrelevance appears similar to _concept shift_, which can drive a specialization-generalization transition; see, eg, arXiv:2409.15582. What is the connection between _prior forgetting & in-context overfitting_ and this concept-shift-induced transition?
- The paper title, _"Prior Forgetting and In-Context Overfitting"_, suggests that this ICL behavior is something we want to avoid, but I understand that it is just another mode of ICL which is sometimes desirable (even though it increases generalization gaps in the settings considered in this paper). Do I miss something important that the title is supposed to convey?
- What are the lessons from this paper that might translate to practical settings? I understand that this paper is purely theoretical, but a discussion of the potential impact (beyond revealing and characterizing interesting phenomena, which I appreciate) would significantly strengthen the paper.
- The training plateau is quite common in ICL training but seems absent from the current analysis. Can the authors comment on this absence?

**Ethical Concerns:**

["NO or VERY MINOR ethics concerns only"]

**Final Justification:**

The authors have sufficiently addressed my questions and clarified my points of confusion. I maintain my rating.

**Limitations:**

yes

**Paper Formatting Concerns:**

No paper formatting concerns

**Quality:**

3

**Strengths And Weaknesses:**

**Strengths**
- A simple model allows for a controlled study of ICL behavior, generating insights that have the potential to generalize to more complex settings
- A simple theory captures task recognition vs task learning competition during pretraining
- The paper is well-structured and mostly clear
- Derivations are easy to follow

**Weaknesses**
- Lack of comparison with benchmark algorithms for linear regression like ridgeless regression and/or Bayes optimal for the noncentred prior consider in the paper; it would be instructive to understand when ICL outperforms naive ridgeless regression by specializing to prior and whether prior forgetting makes ICL solution closer to ridgeless regression (which is optimal for centred isotropic prior)
- The experiments are limited to linear attention; it would be good to see the results validated, even qualitatively, in more realistic settings
- If accepted, the authors should consider using the extra page to explain and discuss what their insights mean for more practical settings; do we see a transition from task recognition to task learning during pretraining in these settings?

---

> ### Author Rebuttal · Authors · 2025-07-27
>
> > Lack of comparison with benchmark algorithms for linear regression like ridgeless regression and/or Bayes optimal for the noncentred prior consider in the paper; it would be instructive to understand when ICL outperforms naive ridgeless regression by specializing to prior and whether prior forgetting makes ICL solution closer to ridgeless regression (which is optimal for centred isotropic prior)
>
> [A1] We are not sure whether we understand your point and would appreciate clarification if we have misinterpreted it.
>
> - Ridgeless regression refers to solving least squares in the overparameterized regime ($d>n$) by selecting the minimum-norm solution. However, if $d>n$, then $n$ demonstrations $\\{(x^{(i)},w^\top x^{(i)})\\}_{i=1}^n$ are not enough to determine the true solution $w$. In the paper, we assume $n\ge d$ and we will clarify this point in the revised version.
>
> - Note that our prior is not designed to pick a Bayes optimal solution among many possible solution, but to model the task distribution. There is only one unique solution when given $n\ge d$ demonstrations.
>
> - If we use least squares with $n>d$, then we can always find the **unique** optimal solution with zero training loss.
> However, our single-layer Transformers are not expressive enough to solve the least squares and cannot achieve zero training loss ($L^*\_{\text{train}}\neq 0$).
>
> ---
> > The experiments are limited to linear attention; it would be good to see the results validated, even qualitatively, in more realistic settings. If accepted, the authors should consider using the extra page to explain and discuss what their insights mean for more practical settings; do we see a transition from task recognition to task learning during pretraining in these settings?
>
> [A2] Thank you for the suggestions. Relaxing the linear attention setting, we trained a more realistic **multi-head ($h=8$) multi-layer ($\ell\ge12$) Transformer with softmax activation and residual connection with larger $n\ge256$ and about a billion training tokens ($\ge$ 1B)** (we used AdamW to accelerate the optimization), and  **observed in-context overfitting phenomenon** similar to what is shown in Fig 3 ($L_{\text{test}}$ decreases and then increases).
> For the general Transformer, it is hard to analyze the dynamics of $\theta(t)$ as we’ve done in the paper with $[\alpha(t),\kappa(t)]$.
>
> We will use the extra page to further elaborate on practical settings in the revised version.
>
> ---
> > In Fig 3, the generalization gaps for two and full-parameter attentions are very similar. Is this a coincidence, or are there reasons to expect this result?
>
> [A3] Yes, the generalization gaps for two- and full-parameter attentions are very similar.
> This similarity arises because
> the full-parameter representation
> $$P=\begin{bmatrix}0_{d\times d}&0_d\\\\ p^\top&\kappa\end{bmatrix},Q=\begin{bmatrix} \bar Q&0_d\\\\ q^\top&0\end{bmatrix}$$
> tends to **converge to the parameters similar to the two-parameter representation**
> $$P=\begin{bmatrix}0_{d\times d}&0_d\\\\ \alpha\mu^\top&\kappa\end{bmatrix},Q=\begin{bmatrix}I_d&0_d\\\\ 0_d^\top&0\end{bmatrix}$$
> up to scaling. A precise analytic explanation remains an open question. We will include the visualization of learned weight for the full-parameter training.
>
> ---
> > The demonstration-query task irrelevance appears similar to concept shift, which can drive a specialization-generalization transition; see, eg, arXiv:2409.15582. What is the connection between prior forgetting & in-context overfitting and this concept-shift-induced transition?
>
> [A4] Thank you for the reference. Its settings seem similar to ours ($w(=\beta),w_q(=\tilde\beta)\sim\mathcal{D}\_\mathcal{W}$).
>
> - We can also validate the main claim "More data hurts performance when concept shift is strong" in [c] from our $L_{\text{test}}^\*$:
> $$L_{\text{test}}^\*=\frac{d+1}{n+d+1}(1+b)+\frac{n}{n+d+1}2b$$
> $\rightarrow$ more data (larger $n$; more weight on $2b$) hurts ($2b>1+b$) if shift is strong ($b$ is large; $b>1$)
>
> ||D-Q task irrelevance|concept shift [c]|
> |-|-|-|
> |task distribution|noncentral|central|
> |focus |training dynamics|performance of pretrained model|
> |$d,n$|non-asymp.|asymptotic ($d,n\rightarrow\infty,d/n\rightarrow\gamma$)|
> |shift measure|task dispersion $b$ of task distribution|$\kappa,\theta$ (pairwise; $\beta\leftrightarrow\tilde\beta$)|
>
> - We also checked other papers [a,b] on specialization-generalization transition.
> We focus on **the pretraining dynamics**, i.e., we draw graphs of parameter ($\theta$) or performance ($L$) vs training steps ($t$), while [a,b] focus on the test performance affected by **the diversity of the pretraining task distribution measured by $\phi$**, i.e., they draw graphs of performance (test error) for a range of $\phi$.
>
> ||prior forgetting| specialization-generalization transition [a,b]|
> |-|-|-|
> |setting (train) |$w=w_q\sim\mathcal{D}\_\mathcal{W}$ |$w=w_q\sim\mathcal{D}\_\mathcal{W}$|
> |setting (test) |$w, w_q\sim_{i.i.d.}\mathcal{D}\_\mathcal{W}$ (independent vectors from the same $\mathcal{D}\_\mathcal{W}$) |$w=w_q\sim\mathcal{D}\_\mathcal{W}{\color{red}'}$ (same vector, but the test task distribution is shifted: $\mathcal{D}\_\mathcal{W}\rightarrow \mathcal{D}\_\mathcal{W}{\color{red}'}$)|
> |focus |training dynamics|performance of pretrained model|
> |phenomena|TR$\downarrow$ and $\vert\kappa\vert\downarrow$ as $t\uparrow$ in the later phase | Test error $\downarrow$ as $\phi\uparrow$ (especailly, $\phi\geq 90^\circ$)|
> |indep. variable|training time ($t$)|task diversity ($\phi$)|
> |dep. variable|performance ($L_{\text{test}},L_{\text{train}},\Delta L$) and parameter ($\theta$)|performance (test error)|
>
> [a] Goddard et al., Specialization-generalization transition in exemplar-based in-context learning (NeurIPSW 2024)
>
> [b] Goddard et al., When can in-context learning generalize out of task distribution? (ICML 2025)
>
> [c] arXiv:2409.15582
>
> ---
> > The paper title, "Prior Forgetting and In-Context Overfitting", suggests that this ICL behavior is something we want to avoid, but I understand that it is just another mode of ICL which is sometimes desirable (even though it increases generalization gaps in the settings considered in this paper). Do I miss something important that the title is supposed to convey?
>
> [A5] Task recognition (TR) and task learning (TL) abilities are both desirable.
> The thing is that, in the later phase of training, the TR ability gets **weaker** (prior forgetting).
> You're right that weak TR ability and a large generalization gap in our setting does **not negatively affect the training risk** (actually the training risk keeps decreasing).
> However, we still want to avoid this behavior, especially when given query task requires prior knowledge or the demonstration is not very informative for the task.
> The statement "in-context overfitting = increasing TL ability" is overly simplified (it is always better with the higher TL ability). We will clarify this with "in-context overfitting = the two ICL abilities (TR and TL) becomes imbalanced and the model becomes overly reliant on the demonstration as TR gets weaker" (Yes, this still does not negatively affect the training risk).
>
> |phase|prior strength $\vert\alpha\vert$|in-context strength $\vert\kappa\vert$|TR|TL|
> |-|-|-|-|-|
> |early|$\vert\alpha\vert{\uparrow}$ (prior learning)|$\vert\kappa\vert{\uparrow}$ (in-context fitting)|TR$\uparrow$|TL$\uparrow$|
> |later|$\vert\alpha\vert{\color{red}\downarrow}$ (prior ${\color{red}\text{forgetting}}$)|$\vert\kappa\vert{\uparrow}$ (in-context ${\color{red}\text{over}}$fitting)|TR${\color{red}\downarrow}$|TL$\uparrow$|
>
> ---
> > What are the lessons from this paper that might translate to practical settings? I understand that this paper is purely theoretical, but a discussion of the potential impact (beyond revealing and characterizing interesting phenomena, which I appreciate) would significantly strengthen the paper.
>
> [A6]
> We can imagine many possible examples due to the simplicity of the analysis.
> For example, based on the understanding, ...
> - we may build a new objective or regularizer (e.g., early stopping, penalty on a small $|\alpha|$) that mitigates the prior forgetting.
> - we may understand a failure mode of LLMs, especially when irrelevant context is given, e.g., Shi et al. (2023).
> - we may design a method to predict the generalization performance of LLMs by monitoring the parameters.
>
> ---
> > The training plateau is quite common in ICL training but seems absent from the current analysis. Can the authors comment on this absence?
>
> [A7] We also observed a training plateau (before abrupt loss drop) when we used **SGD** (instead of Adam or AdamW) to train the **full-parameter** Transformer.
>
> This plateau is predicted by the theory of layerwise linear neural network [d,e].
> Especially, [e] shows that single-layer linear transformer is equivalent to the two-layer linear neural network (LNN) with cubic feature input.
> Let's consider the simplest setup: we want to minimize $L(a,b)=(1-bax)^2$ where $a,b$ represent the parameters of each layer in the two-layer LNN, $x$ the input, and 1 the target ($x\mapsto ax\mapsto bax$). Then, $\dot a =-\frac{\partial L}{\partial a}=2(1-abx)bx$ and $\dot b=-\frac{\partial L}{\partial b}=2(1-abx)ax$. Thus, $c=ab$
> follows the dynamics $\dot c = a\dot b+ b\dot a = 4(1-cx)cx$ which has a sigmoidal solution of $c(t)=\frac{1}{x(C_1\exp(-4xt)+1)}$. Thus, the loss starts with a (long) plateau, exhibits a sudden drop, and saturates. This intuition also holds for general LNNs.
>
> Note that we didn't use SGD to train the full-parameter transformer in Fig 3.
> We will add the corresponding plots that showed a plateau (by using SGD to train the full-parameter Transformer), together with the above discussion.
>
> [d] Saxe et al., Exact solutions to the nonlinear dynamics of learning in deep linear neural networks (ICLR 2014)
>
> [e] Zhang et al., Training Dynamics of In-Context Learning in Linear Attention (ICML 2025)

---

> > ### Comment · Reviewer_25wX · 2025-08-03
> > **Follow-up questions**
> >
> > Thank you for your detailed reply. I have a few follow-up questions.
> >
> > -  **Benchmark algorithms for linear regression.** Thank you for clarifying the important condition on *n* and *d*. I believe this assumption was not in the original paper(?), and if it was, I would appreciate a more prominent placement.
> >     -  What's stopping the analysis of the case *n<d*? The importance of prior is particularly strong in this case. How does it affect "in-context overfitting"?
> >     -  Currently, the tasks are deterministic linear maps. What would be the effects of noisy maps?
> >
> >
> > - **Two and full-parameter attentions similarity.** I find the convergence quite interesting. Has this phenomenon been observed or described before? And is it an artifact of the specific setting considered in this work?
> >
> >
> > - **About the title 'In-Context Overfitting'.** Thanks for clarifying. I understand that both TR and TL are desirable, depending on the task. Now I feel the suggested edit _"the two ICL abilities (TR and TL) becomes imbalanced"_ is a bit vague. What determines whether TR and TL are balanced? Do you think there is a fundamental tradeoff between these two abilities? (Of course, both abilities can improve in the early phase since the model is far from the Pareto frontier.) I would appreciate a discussion about this in the manuscript.
> >
> >
> > -  **Training plateau**
> >     -  _"observed a training plateau (before abrupt loss drop) when we used SGD (instead of Adam or AdamW)"_  Does this mean no plateau for Adam or AdamW? If so, it would be quite surprising. As far as I understand, the training plateau is quite generic for ICL (see, eg, arXiv:2209.11895 and arXiv:2312.03002).
> >
> >     - The theoretical results rely on gradient flow instead of (SGD or the variants used to train transformers) and the model does not capture the training plateau, which is a common, distinct feature of ICL training dynamics. How do you think these limitations affect the generalizability of the insights from this work? At the very least, these points are worth discussing in the paper.

---

> > > ### Author Response · Authors · 2025-08-04
> > >
> > > >Benchmark algorithms for linear regression. (...)
> > >
> > > [A'1-1]
> > > In the paper, we focus on the least square problem and the $n\geq d$ setting, rather than the ridgeless regression and the underdetermined problem ($n<d$), but our analysis still works and exhibits the in-context overfitting even when $n<d$. Now we somewhat understand what you meant in the initial review. We will provide an answer to this one in a separate comment due to the character limit.
> > >
> > > >What would be the effects of noisy maps?
> > >
> > > [A'1-2] Thank you for the suggestion! Noisy maps help mitigate in-context overfitting.
> > >
> > > If we use a noisy map $y^{(i)}=w^\top x^{(i)}+\epsilon^{(i)}$ with $\epsilon^{(i)}\sim_{iid}\mathcal{N}(0,\sigma_\epsilon^2)$ and $E=[\epsilon^{(1)},\cdots,\epsilon^{(n)}]$, then we have noisy $Z^\epsilon=\begin{bmatrix}\bar X&x^{(n+1)}\\\\ \bar Y+E&0\end{bmatrix}$ and
> > > $$T_{P,Q}(Z^\epsilon)=-\frac{1}{n}x^{(n+1)\top}\left[\bar Q^\top \bar X\bar X^\top p +\kappa\bar Q^\top\bar X(\bar Y+E)^\top +q(\bar Y+E)\bar X^\top p+\kappa q(\bar Y+E)(\bar Y+E)^\top\right]$$
> > > $$=x^{(n+1)\top}\left[-G_x(\alpha\mu+\kappa w){\color{red}-\frac{1}{n}\kappa\bar XE^\top}\right].\ (\text{the two-param. setting})$$
> > > Therefore, the "noisy" training loss is
> > > $$L^\epsilon_{\text{train}}=\mathbb{E}\_{w,X,E,\epsilon^{(n+1)}}\left[\left(w^\top x^{(n+1)}+{\color{blue}\epsilon^{(n+1)}} -T_{P,Q}(Z^\epsilon) \right)^2\right]=L_{\text{train}}+{\color{blue}\sigma_\epsilon^2}+{\color{red}\frac{1}{n}\sigma_\epsilon^2\kappa^2}$$
> > > and it has an additional affect to regularize the in-context strength $\kappa^2$ and mitigate the in-context overfitting.
> > >
> > > We empirically check the regularization effect when $\sigma_\epsilon^2$ is large ($\ge 1$).
> > >
> > > ---
> > > >Two and full-parameter attentions similarity. I find the convergence quite interesting. Has this phenomenon been observed or described before? And is it an artifact of the specific setting considered in this work?
> > >
> > > [A'2] We don't know any work on the two and full-parameter attentions similarity, but the transience of ICL (similar to the prior forgetting) has been empirically observed in previous studies (e.g, [a] for a synthetic Omniglot classification tasks and [b] for a Markov mixture tasks).
> > >
> > > [a] Singh et al., The transient nature of emergent in-context learning in transformers (NeurIPS 2023)
> > >
> > > [b] Park et al., Competition dynamics shape algorithmic phases of in-context learning (ICLR 2025)
> > >
> > > ---
> > > >[About the title 'In-Context Overfitting']. Thanks for clarifying. I understand that both TR and TL are desirable, depending on the task. Now I feel the suggested edit "the two ICL abilities (TR and TL) becomes imbalanced" is a bit vague. What determines whether TR and TL are balanced?
> > >
> > > [A'3-1] We can use $|\alpha|$ and $|\kappa|$ values to determine whether TR and TL balanced.
> > > It is fair to say that they becomes imbalanced when $|\kappa(t)|$ keeps increasing but $|\alpha(t)|$ decreases (Fig 2).
> > >
> > > >Do you think there is a fundamental tradeoff between these two abilities?
> > >
> > > [A'3-2] No, we think the two abilities can improve without tradeoff but it requires another training objective (e.g., [A'1-2]).
> > >
> > > ---
> > > >[Training plateau]
> > > The theoretical results rely on gradient flow instead of (SGD or the variants used to train transformers) and the model does not capture the training plateau, which is a common, distinct feature of ICL training dynamics. How do you think these limitations affect the generalizability of the insights from this work? At the very least, these points are worth discussing in the paper.
> > >
> > > >"observed a training plateau (before abrupt loss drop) when we used SGD (instead of Adam or AdamW)" Does this mean no plateau for Adam or AdamW? If so, it would be quite surprising. As far as I understand, the training plateau is quite generic for ICL (see, eg, arXiv:2209.11895 and arXiv:2312.03002).
> > >
> > > [A'4]
> > > - (SGD vs Adam/AdamW) Adam or AdamW also show plateaus, but they remain very short compared to SGD (see steps $\sim 10$ in Fig 3 (Bottom)). We compare SGD and AdamW ($\text{lr}=0.01, n=10, d=5, \sigma=0.4$) with weight initialization scale of 1e-4.
> > > ||two-param Transformer| full-param single layer Transformer|softmax-based 3-layer Transformer with residual connection|
> > > |---|---|---|---|
> > > |SGD|no plateau|long plateau ($t\times\text{lr}\sim 1$)|long plateau ($t\times\text{lr}\sim 15$)|
> > > |Adam/AdamW|no plateau|short plateau ($t\times\text{lr}\sim 0.1$)|short plateau ($t\times\text{lr}\sim 0.1$)|
> > > - Two-param model is limited in capturing the training plateau, but it does explain the u-shape overfitting dynamics well. We will include this limitation in the paper.
> > > - (GF vs SGD) Fig 2 (Top) and 3 (Top) show that SGD (solid lines) and GF (dashed lines) have very similar trajectory and loss curves for two-param model.
> > > - Sometimes ICL does not exhibit plateau when using a large initialization (e.g., Fig 6 in arxiv:2501.16265).

---

> > > ### Author Response · Authors · 2025-08-04
> > >
> > > >Benchmark algorithms for linear regression. Thank you for clarifying the important condition on n and d. I believe this assumption was not in the original paper(?), and if it was, I would appreciate a more prominent placement.
> > > What's stopping the analysis of the case n<d? The importance of prior is particularly strong in this case. How does it affect "in-context overfitting"?
> > >
> > > >[**Original Question**] Lack of comparison with benchmark algorithms for linear regression like ridgeless regression and/or Bayes optimal for the noncentred prior consider in the paper; it would be instructive to understand when ICL outperforms naive ridgeless regression by specializing to prior and whether prior forgetting makes ICL solution closer to ridgeless regression (which is optimal for centred isotropic prior)
> > >
> > > [A'1-1]+[A1]
> > > In the paper, we focus on the least square problem and the $n\geq d$ setting, but our analysis still works and exhibits the in-context overfitting even when $n<d$.
> > > Below we tried to answer the [**Original Question**].
> > >
> > > First, we put $P_x=\bar X(\bar X^\top \bar X)^{-1}\bar X^\top$ for $n<d$.
> > > ||$n,d$|solution|$\arg\min$|
> > > |-|-|-|-|
> > > |least squares |$n\geq d$| $w_{ls}=(\bar X\bar X^\top)^{-1}\bar X \bar Y^\top$|$=\arg\min_w \Vert\bar X^\top w-\bar Y^\top\Vert^2$ |
> > > |ridgeless regression|$n<d$|$w_{rl}=P_x w$|$=\arg\min_{\bar X^\top w=\bar Y^\top}\Vert w\Vert^2$|
> > > |Bayes optimal |$n<d$| $w_{b}=P_x (w-\mu) +\mu$|$=\arg\min_{\bar X^\top w=Y^\top}\Vert w-\mu\Vert^2$|
> > >
> > > We can compute the Bayes optimal loss $L_b$, ridgeless loss $L_{rl}$, and compare it with our optimal training loss $L_{\text{train}}^\ast$.
> > > - $$L_{b}=\mathbb{E}[(w_{b}^\top x^{(n+1)}-w^\top x^{(n+1)})^2]=\mathbb{E}[(w_{b}-w)^\top x^{(n+1)})^2]=\mathbb{E}[\\|w_{b}-w\\|^2]$$
> > > $$=\mathbb{E}[\\|P_x(w-\mu)+\mu-w\\|^2]=\mathbb{E}[\\|(P_x-I_d)(w-\mu)\\|^2]$$
> > > $$=\mathbb{E}[\text{Tr}((P_x-I_d)^2(w-\mu)(w-\mu)^\top)]=\mathbb{E}[\text{Tr}((I_d-P_x)\sigma^2 I_d)]$$
> > > $$\color{red}{=(d-n)\sigma^2}$$
> > >
> > > - $$L_{rl}=\mathbb{E}[(w_{rl}^\top x^{(n+1)}-w^\top x^{(n+1)})^2]=\mathbb{E}[\text{Tr}((P_x-I_d)^2ww^\top)]=\mathbb{E}[\text{Tr}((I_d-P_x)(\mu\mu^\top+\sigma^2 I_d))]$$
> > > $$\color{red}{=(d-n)\left(\frac{1}{d}+\sigma^2\right)}$$
> > > $$\color{blue}{=\frac{d-n}{d}(1+d\sigma^2)}$$
> > >
> > > - $$L_{\text{train}}^\ast = {\color{blue}{\frac{d+1}{n+d+1}(1+d\sigma^2)}}$$
> > >
> > > We can easily see that $\color{blue}{\frac{d-n}{d}<\frac{d+1}{n+d+1}}$ and thus $L_b\ {\color{red}<}\ L_{rl}\ {\color{blue}<}\ L_{\text{train}}^\ast$.
> > > ICL (with the two-parameter model) never outperforms the ridgeless regression.
> > > Empirically, we observe that 12-layer Transformer achieves a similar performance with the ridgeless solution. This may be related to the results "Transformers learn in-context by gradient descent" [1] and "GD has implicit bias toward the minimum norm solution" [2].
> > >
> > > ---
> > > [1] von Oswald et al., Transformers learn in-context by gradient descent (ICML 2023)
> > >
> > > [2] Gunasekar et al., Implicit Regularization in Matrix Factorization (NeurIPS 2017)

---

> > > > ### Comment · Reviewer_25wX · 2025-08-06
> > > >
> > > > Thank you for the response. I appreciate the discussion. It helps me understand the results better. I don't have further questions. I will maintain my positive rating.

---

### Official Review · Reviewer_NtKW · 2025-06-26

**Clarity:** 2
**Significance:** 2
**Originality:** 3
**Rating:** 3
**Confidence:** 4

**Summary:**

This paper investigates the training dynamics of ICL with transformer, with simplified linear transformer and synthetic linear regression tasks. To decouple and evaluate the task recognition and task learning ability, this paper propose to investigate with demonstration-query task irrelevance. Results mainly show that the model first learns the task learning and the task recognition abilities together in the beginning, but it gradually forgets the task recognition ability to recall the priorly learned tasks and relies more on the given context in the later phase.

**Questions:**

Q1. With some specific settings in this paper, will it be possible to directly identify the global minimum (like [3]), thus illustrate the main results as the process of parameter convergence?

Q2. Why not train in auto-regressive manner? And will the context length matter the main results?

[3] Ahn K, Cheng X, Daneshmand H, et al. Transformers learn to implement preconditioned gradient descent for in-context learning[J]. Advances in Neural Information Processing Systems, 2023, 36: 45614-45650.

**Ethical Concerns:**

["NO or VERY MINOR ethics concerns only"]

**Final Justification:**

This is a technically solid paper. But some settings and assumptions, and how closely they are related to real-world LLMs, to be valuable, worth further discussion.

**Quality:**

2

**Strengths And Weaknesses:**

The topic is interesting and important. Understanding ICL, from early works which focus on how a learning algorithm can be expressed, to recent works (including this paper) which focus on how the ability emerges from pre-training, helps us to understand, explain and improve LLMs.

The first noticeable (but not that important) weakness is that latest related works about understanding ICL with transformer are not discussed nor mentioned, e.g., [1][2].

The main weakness of this paper is critical: the setting has departed too far from real world LLMs to be meaningful. Three levels of departures can be recognized:

First, the linear transformer and synthetic linear regression tasks. Though different from real LLMs, the reviewer can accept this one in most cases, as it facilitates research.

Second, pre-training with explicitly arranged ICL input ([xy,xy,x?]). The reviewer think this is not  to investigate how the ICL ability emerges from pre-training. Because this is not reduction problem any more, as modern LLMs do not pre-train in such cases.

Third, some unique settings in this paper. The reviewer could not understand what the value of loss between an random label that is independent with input means (Demonstration-Query Task Irrelevance). And the mean value of linear weights is specific to linear regression task (quadratic loss), while LLMs do pre-training neither with such tasks nor with such loss.

Based on above judgement, the reviewer has no longer check the result and proof details.

[1] Wu S, Wang Y, Yao Q. Why In-Context Learning Models are Good Few-Shot Learners?[C]//The Thirteenth International Conference on Learning Representations.
[2] Yang L, Lin Z, Lee K, et al. Task Vectors in In-Context Learning: Emergence, Formation, and Benefit[J]. arXiv preprint arXiv:2501.09240, 2025.

---

> ### Author Rebuttal · Authors · 2025-07-27
>
> > The first noticeable (but not that important) weakness is that latest related works about understanding ICL with transformer are not discussed nor mentioned, e.g., [1][2].
>
> [A1] Thank you for the suggestions. We will include **more discussion** on latest related works (e.g., [1], [2]).
>
> ---
> > First, the linear transformer and synthetic linear regression tasks. Though different from real LLMs, the reviewer can accept this one in most cases, as it facilitates research.
>
> [A2] Yes, there has been extensive research exploring the **linear transformer** [Ahn et al., 2024a,b, Dai et al., 2023a, Mahankali et al., 2024, Sander et al., 2024, Schlag et al., 2021, Von Oswald et al., 2023a, Von Oswald et al., 2023b, Zhang et al., 2021, Zheng et al., 2024] and the **in-context linear regression task** [Ahn et al., 2024a,b, Akyürek et al., 2023, Garg et al., 2022, Li et al., 2023a, Li et al., 2023b, Lin and Lee, 2024, Mahankali et al., 2024, Raventós et al., 2024, Von Oswald et al., 2023a, Zhang et al., 2024].
>
> Relaxing the linear attention setting, we also trained a more realistic **multi-head ($h=8$) multi-layer ($\ell\geq12$) Transformer with softmax activation and residual connection with larger $n\geq 256$ and about a billion training tokens ($\geq$ 1B)** (we used AdamW to accelerate the optimization), and **observed in-context overfitting phenomenon** similar to what is shown in Figure 3 ($L_{\text{test}}$ decreases and then increases). We will provide more discussion for the practical settings (including the above) in the revised version.
>
> ---
> > Second, pre-training with explicitly arranged ICL input ([xy,xy,x?]). The reviewer think this is not to investigate how the ICL ability emerges from pre-training. Because this is not reduction problem any more, as modern LLMs do not pre-train in such cases.
>
> [A3] The objective we (and many others) considered is designed to model the **next-token prediction objective** typically used to pretrain modern LLMs.
>
> - We (and many others [Ahn et al., 2024a,b, Mahankali et al., 2024, Von Oswald et al., 2023a, Zhang et al., 2024]) use the prompt $$Z=\begin{bmatrix}x^{(1)} &\cdots &x^{(n)} &x^{(n+1)}\\\\ f(x^{(1)}) &\cdots &f(x^{(n)}) &0 \end{bmatrix}\in\mathbb{R}^{(d+1)\times (n+1)}$$
> to treat $(x,y)$ as a combined context token to simplify the analysis.
>
> - Garg et al. (2022) use the prompt
> $$Z=\begin{bmatrix} x^{(1)}, &\begin{bmatrix} f(x^{(1)}) \\\\ 0_{d-1}\end{bmatrix}, \cdots, x_n,\begin{bmatrix} f(x^{(n)}) \\\\ 0_{d-1}\end{bmatrix}, x^{(n+1)}\end{bmatrix}\in\mathbb{R}^{d\times (2n+1)}$$
>
> - Akyurek et al. (2023) use the prompt
> $$Z=\begin{bmatrix} x^{(1)} & 0_d & \cdots &x^{(n)} &0_d &x^{(n+1)}\\\\ 0& f(x^{(1)}) &\cdots& 0 & f(x^{(n)}) &0 \end{bmatrix} \in \mathbb{R}^{(d+1)\times (2n+1)}$$
>
> ---
> > Third, some unique settings in this paper. The reviewer could not understand what (i) the value of loss between (ii) an random label that is independent with (iii) input means (Demonstration-Query Task Irrelevance). And (iv) the mean value of linear weights is specific to linear regression task (quadratic loss), while LLMs do pre-training neither with such tasks nor with such loss.
>
> [A4] (i) $L\_{\text{test}} = \mathbb{E}[||w\_q-\hat w||^2]$, (ii) $w_q$, (iii) $w$, (iv) $\mu$
>
> Our goal is to **disentangle** the strengths of the task recognition (TR) and task learning (TL) abilities and quantitatively analyze them **separately**.
>
> - The usual setting is to use the same query task as the demonstration task ($w_q=w$) just like our $L_{\text{train}}$. Then we cannot isolate and quantify each effect independently (**TL+TR**).
> - However, if we use the query task **different** from the demonstration ($w_q\neq w$) just like our $L_{\text{test}}$, then we can measure how much the model relies on the prior (**TR**) as it cannot rely on the demonstration.
>
> In other words, ...
> - [phase 1] $L_{\text{train}}$ and $L_{\text{test}}$ are both good (small). This indicates that the model relies not only on the demonstration, but also on the prior (described by the mean vector $\mu$) that both $w_q=\mu+s_q$ and $w=\mu+s$ share  (**TL+TR**).
> - [phase 2] $L_{\text{train}}$ is good but $L_{\text{test}}$ is not. This indicates that the model relies more on the demonstration (**TL**).
>
> Our main results show that
> - In the early stage of training [phase 1], the model learns (i) the task prior distribution [TR; increasing prior strength $|\alpha|$] and (ii) how to **fit** to the demonstration data [TL; increasing in-context strength $|\kappa|$] **(TR+TL $\uparrow$)**.
> - In the later phase [phase 2], as $|\alpha|$ decreases (prior forgetting), the two ICL abilities (TR and TL) **becomes imbalanced**, and the model becomes **overly** reliant on the demonstration, which we call in-context **over**fitting **(TR $\downarrow$, TL $\uparrow$)**.
>
> | phase | prior strength $\vert\alpha\vert$ | in-context strength $\vert\kappa\vert$ | TR | TL |
> |---|---|---|---|---|
> | early phase | $\vert\alpha\vert{\uparrow}$ (prior learning) | $\vert\kappa\vert{\uparrow}$ (in-context fitting)| TR$\uparrow$ | TL$\uparrow$ |
> | later phase | $\vert\alpha\vert{\color{red}\downarrow}$ (prior ${\color{red}\text{forgetting}}$) | $\vert\kappa\vert{\uparrow}$ (in-context ${\color{red}\text{over}}$fitting) |TR${\color{red}\downarrow}$ |TL$\uparrow$ |
>
>
> ---
> > Q1. With some specific settings in this paper, will it be possible to directly identify the global minimum (like [3]), thus illustrate the main results as the process of parameter convergence?
>
> [A5] Yes, with our specific settings, we are able to directly identify the global minimum in [3] **as a special case when $\mu=0$**.
>
> With $\mu=0$, our parameterization $P=\begin{bmatrix} 0_{d\times d}& 0_d\\\\ 0_d^\top& \kappa \end{bmatrix}$ and $Q=\begin{bmatrix} I_{d}& 0_d\\\\ 0_d^\top& 0\end{bmatrix}$ (see eq (6)) yields the convergence point (see eq (9)):
> $$P^\ast=\begin{bmatrix} 0_{d\times d}& 0_d\\\\ 0_d^\top& \kappa^\ast \end{bmatrix}, Q^\ast=\begin{bmatrix} I_{d}& 0_d\\\\ 0_d^\top& 0\end{bmatrix}\\ \text{with}\\ \kappa^\ast=-\frac{n}{n+d+1}$$
> This is equivalent to the following, since $P,Q$ represent the same model up to rescaling $P\leftarrow \gamma P$ and $Q\leftarrow \gamma^{-1}Q$ for a scalar $\gamma$:
>
> - [A] **Our global minimum** ($\mu=0$)
> $$P^\ast=\begin{bmatrix} 0_{d\times d}& 0_d\\\\ 0_d^\top& 1 \end{bmatrix}, Q^\ast=\begin{bmatrix} {\color{red}{\kappa^\ast}} I_{d}& 0_d\\\\ 0_d^\top& 0\end{bmatrix} \text{with}\ \color{red}{\kappa^*=-\frac{n}{n+d+1}}$$
> This reduces to Theorem 1 in Ahn et al. (2024a) and their equation (7) when $\Sigma=I$ is shown below:
>
> - [B] **Global minimum in Ahn et al. (2024a) ($\Sigma=I$)**
> $$P^\ast=\begin{bmatrix} 0_{d\times d}& 0_d\\\\ 0_d^\top& 1 \end{bmatrix}, Q^\ast=-\frac{1}{\frac{n-1}{n}+(d+2)\frac{1}{n}}\begin{bmatrix} I_{d}& 0_d\\\\ 0_d^\top& 0\end{bmatrix}=\begin{bmatrix} {\color{red}{-\frac{n}{n+d+1}}} I_{d}& 0_d\\\\ 0_d^\top& 0\end{bmatrix}$$
>
> - Therefore, we have **[A]=[B]**. One more thing we want to emphasize is that, using a (relatively) general parameterization,  Ahn et al. (2024a) can only derive the global optimum. On the other hand, using a simple parameterization, we can derive the exact dynamics of pretraining and obtain the convergence point (optimum).
>
> ---
> > Q2-1. Why not train in auto-regressive manner?
>
> [A6]
> As answered in [A3], Garg et al. (2022) and Akyurek et al. (2023) train the model in auto-regressive manner. Similarly, the objective we (and many others) used also aims to model the next-token prediction objective typically used to pretrain modern LLMs, but we treat $(x,y)$ as a **combined context token** to simplify the analysis.
>
> ---
> > Q2-2. And will the context length matter the main results?
>
> [A7] Theorem 4.1-4.2 and Figure 2-3 show that the task dispersion $b=\mathrm{tr}(\Sigma_{\mathcal{W}})=\sigma^2 d$ plays an important role in determining the objective $L_{\text{train}}(\theta)$ and thus the dynamics $\theta(t)$. The context length $n$ also matters the main results, but does not affect the **trajectory shape** of the dynamics.
> - We can rewrite eq (10) as follows:
> $$\theta(t)=c*\underbracket{ (e^{-\lambda\_+t}v\_+-e^{-\lambda\_-t}v\_-+[0,\sqrt{4+b^2}]^\top) }\_{\text{only depends on $b$, not on $n$}}=c*f(t;b),\ \text{where the scaling factor}\ c=\frac{n}{n+d+1}\frac{1}{\sqrt{4+b^2}}$$
> since $\lambda_{\pm}$ and $v\_\pm$ only depend on $b$, not $n$.
> The context length $n$ can only **scale** the dynamics and does not affect the **shape** of the dynamics. Even in the case of $n\gg d$, the scaling factor $c\approx \frac{1}{\sqrt{4+b^2}}$ is almost constant with respect to $n$, and $n$ has little to no effect to control the scale.

---

> > ### Comment · Reviewer_NtKW · 2025-08-07
> > **Thanks for your reponse**
> >
> > Thank the authors for the detailed rebuttal. Some of my concerns have been addressed. This is a technically solid paper. I will raise my rating.
> > But some settings and assumptions, and how closely they are related to real-world LLMs, to be valuable, worth further discussion.

---

### Official Review · Reviewer_EWen · 2025-07-03

**Clarity:** 3
**Significance:** 4
**Originality:** 3
**Rating:** 5
**Confidence:** 3

**Summary:**

This paper focuses on the in-context learning (ICL) ability of large language models (LLMs), particularly during the pretraining phase. The authors investigate the emergence and dynamics of two learning modes, task recognition and task learning. It also reveals two key phenomena: Prior Forgetting and In-Context Overfitting.

**Questions:**

1. Based on Figure 1 (left) and the definition of in-context overfitting in lines 250–251, there seems to be a one-to-one correspondence between increasing in-context strength and overfitting. However, in the early stage of training, models are typically underfitting. How should this apparent contradiction be understood?
2. Is there a connection between the minimum point of the $L_{test}$ curve and the minimum of $\alpha(t)$? If so, could this be explained or analyzed more clearly?
3. Why do Figures 2 and 3 not use the same task dispersion setting? Using a consistent setup would help make comparisons more interpretable.
4. Can the findings in this paper offer any practical guidance for improving pretraining strategies or designing better ICL prompts?

**Ethical Concerns:**

["NO or VERY MINOR ethics concerns only"]

**Final Justification:**

My concerns have been addressed. I maintain my rating.

**Limitations:**

No, although limitations are mentioned in the checklist, they are not clearly explained in the main text.

**Quality:**

3

**Strengths And Weaknesses:**

**Strengths:**

- **Originality:** The study explores ICL dynamics from a pretraining perspective, which provides a fresh and insightful perspective. The identification of Prior Forgetting and In-Context Overfitting as emergent phenomena is novel and thought-provoking.

- **Significance:** Understanding how ICL abilities emerge during pretraining can have a substantial impact on the future design of language models, especially regarding pretraining efficiency and generalization capabilities.

- **Quality:** The paper includes thorough and detailed analysis and experiments, which help substantiate the proposed claims and theoretical framing.

- **Clarity:** This paper is well-written and easy to follow. However, there is room for improvements as I mention below.


**Weaknesses:**

1. The proposed framework is conceptually novel, especially in its focus on the emergence and dynamics of task recognition and task learning during pretraining. However, much of the empirical analysis relies on simplified models (e.g., two-parameter or single-layer linear transformers), which may limit the extent to which the framework can be fully validated. This also raises concerns about the generalizability of the findings to large-scale, real-world LLMs.

2. **Clarity**
- 2.1 Figure 1 (left) lacks clarity: the meaning of the axes, curves, and color coding is not well-explained.
- 2.2 Notational consistency can be improved. For example, distinguishing vectors (e.g., $\textbf{x}^{(i)}$ ) from scalars (e.g., $y^{(i)}$) using bold font would enhance readability.

---

> ### Author Rebuttal · Authors · 2025-07-29
>
> >The proposed framework is conceptually novel, especially in its focus on the emergence and dynamics of task recognition and task learning during pretraining. However, much of the empirical analysis relies on simplified models (e.g., two-parameter or single-layer linear transformers), which may limit the extent to which the framework can be fully validated. This also raises concerns about the generalizability of the findings to large-scale, real-world LLMs.
>
> [A1]
> Thank you for the feedback. Beyond the simplified model, we trained a more complex and practical **multi-head ($h=8$) multi-layer ($\ell\geq12$) Transformer with softmax activation and residual connection with larger $n\geq 256$ and about a billion training tokens ($\geq$ 1B)** (we used AdamW to accelerate the optimization) to validate our claim, and **observed in-context overfitting phenomenon** similar to what is shown in Figure 3 ($L_{\text{test}}$ decreases and then increases). We will include an additional discussion in the revised version.
>
> ---
> >2.1 Figure 1 (left) lacks clarity: the meaning of the axes, curves, and color coding is not well-explained.
>
> [A2.1] We will clarify the exact meaning of the axes, curves and color coding in detail.
>
> Figure 1 (Left) and Figure 2 (Bottom, Left) are the same ($\alpha$-$\kappa$ graph), but they are slightly rescaled (we used the same setting; SGD, $n=10,d=5, \sigma=0.2$, etc.).
>
> We intentionally chose not to name the axes (e.g., $\alpha,\kappa$) in Figure 1 saying "decreasing prior strength" and "increasing in-context strength", because Fig 1 is in the introduction section where we didn't yet introduce the definition of $\alpha,\kappa$.
> We already explained the color coding in Figure 1 (the trajectory "from magenta to lime").
>
>
> >2.2 Notational consistency can be improved. For example, distinguishing vectors (e.g., $\mathbf{x}^{(i)}$) from scalars (e.g., $y^{(i)}$) using bold font would enhance readability.
>
> [A2.2] Thank you for the suggestions. We will enhance readability in the revised version.
>
> ---
> >Based on Figure 1 (left) and the definition of in-context overfitting in lines 250–251, there seems to be a one-to-one correspondence between increasing in-context strength and overfitting. However, in the early stage of training, models are typically underfitting. How should this apparent contradiction be understood?
>
> [A3] In the early stage of training, model are not overfitting yet. In-context overfitting **emerges in the later phase**.
>
> - In the early stage of training, the model learns (i) the task prior distribution [task recognition (TR); increasing prior strength $|\alpha|$] and (ii) how to **fit** to the demonstration data [task learning (TL); increasing in-context strength $|\kappa|$].
> - In the later phase, as the prior strength $|\alpha|$ decreases (prior forgetting), the two ICL abilities (TR and TL) **become imbalanced**, and the model becomes **overly** reliant on the demonstration, which we call in-context **over**fitting.
>
> | phase | prior strength $\vert\alpha\vert$ | in-context strength $\vert\kappa\vert$ | TR | TL |
> |---|---|---|---|---|
> | early phase | $\vert\alpha\vert{\uparrow}$ (prior learning) | $\vert\kappa\vert{\uparrow}$ (in-context fitting)| TR$\uparrow$ | TL$\uparrow$ |
> | later phase | $\vert\alpha\vert{\color{red}\downarrow}$ (prior ${\color{red}\text{forgetting}}$) | $\vert\kappa\vert{\uparrow}$ (in-context ${\color{red}\text{over}}$fitting) |TR${\color{red}\downarrow}$ |TL$\uparrow$ |
>
> We will clarify this point by moving the starting point of the downward "In-Context Overfitting" arrow in Figure 1 (left) slightly lower to indicate that in-context overftitting happens in the later phase.
>
> ---
> > Is there a connection between the minimum point of the $L_{\text{test}}$ curve and the minimum of $\alpha(t)$? If so, could this be explained or analyzed more clearly?
>
> [A4] Figure 4 in Appendix E shows the behaviors of $L_{\text{test}}$ and $\alpha(t)$ together.
> As shown in Figure 4, the point when $\alpha(t)$ reaches its minimum does **NOT match** the minimum of $L_{\text{test}}$ on the level set.
>
> Instead, we can provide the following connection between the behaviors of $L_{\text{test}}$ and $\alpha(t)$.
> We have
> $$L_{\text{train}}(\theta)=\frac{1}{2}\theta^\top C_2\theta+C_1^\top\theta+C_0,$$
> $$L_{\text{test}}(\theta)=\frac{1}{2}\theta^\top C_2\theta+{\color{red}{C_1'}}^\top\theta+C_0$$
> where ${\color{red}{C_1'}}=C_1+[2,2]^\top$ and
> $$\frac{dL_{\text{test}} }{d\theta} = C_2\theta+{\color{red}C_1'}=\frac{dL_{\text{train}} }{d\theta}+[2,2]^\top=-\dot\theta+[2,2]^\top$$
> which yields the following:
> $$\frac{dL_{\text{test}} }{dt}=\frac{dL_{\text{test}} }{d\theta}\frac{d\theta}{dt}= (C_2\theta+{\color{red}C_1'})^\top (C_2\theta+C_1)=(\dot\theta-[2,2]^\top)^\top\dot\theta$$
> $$ = \\|\dot\theta^\top-[1,1]^\top\\|^2-2= (\alpha'(t)-1)^2+(\kappa'(t)-1)^2-2$$
> Therefore, if **the condition $(\kappa'(t)-1)^2=1$ (e.g., $\kappa'(t)=0$) holds**, then $\alpha'(t)=0$ implies $\frac{dL_{\text{test}}}{dt}=0$, i.e, $L_{\text{test}}$ is minimized at the minimum point of $\alpha(t)$.
>
> ---
> > Why do Figures 2 and 3 not use the same task dispersion setting? Using a consistent setup would help make comparisons more interpretable.
>
> [A5] Thank you for the suggestion. We will include the plots for $b=0.2,0.6,0.8,3.2$ which cover the settings in the both Figure 2 ($b=0.2,0.8,3.2$) and Figure 3 ($b=0.2,0.6,0.8$).
> We intentionally used different task dispersion settings for each Figure for better visualization. Figure 2 (trajectory dynamics) is less sensitive to $b$ compared to Figure 3 (loss curves). Thus we use a wider range of $b=0.2,0.8,3.2$ for Figure 2, rather than $b=0.2,0.6,0.8$ for Figure 3.
>
> ---
> > Can the findings in this paper offer any practical guidance for improving pretraining strategies or designing better ICL prompts?
>
> [A6]
> We can imagine many possible examples due to the simplicity of the analysis. For example, based on our understanding, ...
> - we may build a new objective or regularizer (e.g., early stopping, penalty on a small $|\alpha|$) that mitigates the prior forgetting.
> - we may understand a failure mode of LLMs, especially when irrelevant context is given, e.g., Shi et al. (2023).
> - we may design a method to predict the generalization performance of LLMs by monitoring the parameters.
>
> As our main goal is to provide a simple analysis, we leave their validation in a practical setting as a future work.

---

> > ### Comment · Reviewer_EWen · 2025-08-05
> >
> > Thank you for the detailed responses. My concerns have been fully addressed. I maintain my original score.

---

### Decision · Program_Chairs · 2025-09-17

**Decision:**

Accept (poster)

**Comment:**

(a) summary: The paper investigates the phenomenon of in-context learning (ICL). Instead of the previously addressed test-time analysis, it focuses on the pre-training phases and uses a simplified linear-attention based transformer and a set of linear-regression tasks to enable theoretical analysis in mathematically tractable form. The analysis looks at the dynamics of task recognition and task learning abilities and reveals two key aspects of the ICL named in the paper prior forgetting and in-context overfitting.

(b) strength: The paper investigates an important phenomenon and puts forward fresh insights with possible impact on future developments in LLMs. The theoretical analysis is solid and thorough and is supported by relevant experiments. The paper is well written and structured.

(c) weaknesses: The simplifying assumptions (linear transformer, linear regression tasks) raise doubts about its validity and generalization to more complex and more realistic set-ups used in real-world LLMs.

(d) reasons to accept: ICL is interesting and important topic for better understanding of LLM behavoiur and abilities. The presented ICL analysis is novel, technically strong and though relying on unrealistically simplified setup provides insights that may serve as a useful stepping stone towards more complex analysis in the future. For these reasons I am leaning to accepting the paper.
At the time some doubts about the generalizability and practical relevance of the presented results to real-life and more complex LLM prevail. I would therefore not oppose if the paper was dumped down.

(e) review discussion: Reviewers appreciated the novel approach to the analysis (pre-training instead of test-time only) and the general importance and hence interest of the in-context learning phenomenon. In addition to asking for some technical clarifications, they expressed their concerns about validity and practical relevance of the simplified set-up in realistic and far more complex LLM pipelines. In addition to the simplified models, the whole pre-training setup of the \emph{demostration-query task irrelevance} has been questioned as inconsequential for any real-life LLM settings. Authors answered the technical questions in the rebuttal to the satisfaction of the reviewers and where appropriate suggested minor clarification updates for the camera-ready. To address the generalization question, they have conducted additional experiment on more complex soft-max based transformer architecture, claiming similar observed behaviours and proposed to cover this in the final version. They also clarified the pre-training setup and its motivation and relevance for actual LLM pre-training. Though some doubts about the generalization of the simplified approach and hence its validity and usefulness for real-life LLMs prevailed, the reviewers were generally satisfied with the rebuttal answers and appreciated the theoretical strengths of the analysis.